# Image Inpainting via Iteratively Decoupled Probabilistic Modeling

**Wenbo Li[1], Xin Yu[2], Kun Zhou[3], Yibing Song[4], Zhe Lin[5]**
[1]Huawei Noah's Ark Lab, [2]HKU, [3]CUHK (SZ), [4]Alibaba DAMO Academy, [5]Adobe Research
{fenglinglwb,yuxin27g,zhoukun303808,yibingsong.cv}@gmail.com

## Abstract

Generative adversarial networks (GANs) have made great success in image inpainting yet still have difficulties tackling large missing regions. In contrast, iterative probabilistic algorithms, such as autoregressive and denoising diffusion models, have to be deployed with massive computing resources for decent effect. To achieve high-quality results with low computational cost, we present a novel pixel spread model (PSM) that iteratively employs decoupled probabilistic modeling, combining the optimization efficiency of GANs with the prediction tractability of probabilistic models. As a result, our model selectively spreads informative pixels throughout the image in a few iterations, largely enhancing the completion quality and efficiency. On multiple benchmarks, we achieve new state-of-the-art performance. Our code and models will be publicly available.

## 1 Introduction

Image inpainting, a fundamental computer vision task, aims to fill the missing regions in an image with visually pleasing and semantically appropriate content. It has been extensively employed in graphics and imaging applications, such as photo restoration Wan et al. (2020; 2022), image editing Barnes et al. (2009); Jo & Park (2019), compositing Levin et al. (2004), re-targeting Cho et al. (2017), and object removal Criminisi et al. (2004). This task, especially filling large holes, is more ill-posed than other restoration problems, necessitating models of stronger generation abilities.

In past years, generative adversarial networks (GANs) have made great processes in image inpainting Pathak et al. (2016); Yan et al. (2018); Yu et al. (2018); Liu et al. (2019); Wan et al. (2021); Li et al. (2022); Chu et al. (2023); Sargsyan et al. (2023). By implicitly modeling a target distribution through a min-max game, GANs-based methods significantly outperform traditional exemplar-based techniques Hays & Efros (2007); Sun et al. (2005); Criminisi et al. (2004; 2003) in terms of visual quality. However, the one-shot generation of GANs sometimes lead to unstable training Salimans et al. (2016); Gulrajani et al. (2017); Kodali et al. (2017) and makes it challenging to learn a complex distribution, particularly when inpainting large holes in high-resolution images.

Conversely, autoregressive models Van den Oord et al. (2016); Van Den Oord et al. (2016); Parmar et al. (2018) and denoising diffusion models Song & Ermon (2019); Ho et al. (2020); Dhariwal & Nichol (2021) recently demonstrated remarkable power in content generation Ramesh et al. (2022); Saharia et al. (2022b); Yu et al. (2022); Singer et al. (2022). These models utilize tractable probabilistic modeling techniques to iteratively refine the image based on prior estimations, resulting in more stable training and improved coverage. However, it is widely known that autoregressive models process images pixel by pixel, which makes it cumbersome to handle high-resolution data. On the other hand, denoising diffusion models typically require thousands of iterations to achieve accurate estimations. Thus, using these methods directly in image inpainting incurs respective drawbacks – *strategies for high-quality large-hole high-resolution image inpainting still fall short*.

To complete the map of inpainting, in this paper, we develop a new pixel spread model (PSM) tailored for the large-hole scenario. PSM operates in an iterative manner, where all pixels are predicted in parallel during each iteration, and only qualified predictions are retained for subsequent iterations. It acts as a process to gradually spread trustful pixels to unknown locations. Our core design lies in a simple yet highly effective decoupled probabilistic modeling (see Section 3.1.1), which enjoys the merits of GANs' efficient optimization and the tractability of probabilistic models. In detail, our

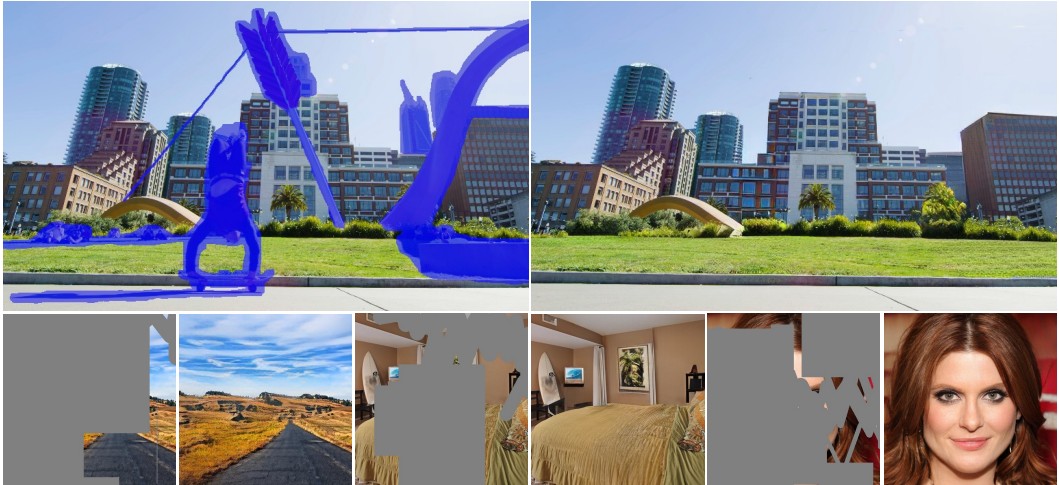

Figure 1: Our model supports photo-realistic large-hole inpainting for various scenarios. The first example for object removal is a high-resolution image captured in the wild, while others ($512 \times 512$) come from Places2 Zhou et al. (2017) and CelebA-HQ Karras et al. (2018) datasets.

model simultaneously predicts an inpainted result (*i.e.*, the mean term) and an uncertainty map (*i.e.*, the variance term). The mean term is optimized using implicit adversarial training, yielding more accurate predictions with fewer iterations. The variance term, contrarily, is modeled explicitly using Gaussian regularization.

The adoption of our decoupled strategy offers numerous advantages. First, the use of adversarial optimization leads to a significant reduction in the number of iterative steps required to achieve promising results, as shown in Figure K.1, much faster than autoregressive and denoising diffusion models. Second, the Gaussian regularization employed produces a variance term that naturally acts as an uncertainty measure (see Section 3.1.2). This allows for the selection of reliable estimates for iterative refinement, largely reducing GANs' artifacts. Furthermore, the explicit modeling of the distribution facilitates continuous sampling, thereby producing predictions with enhanced quality and diversity, as demonstrated in Section 4. Ultimately, the uncertainty measure is instrumental in constructing an uncertainty-guided attention mechanism (see Section 3.2), which encourages the network to leverage more informative pixels for efficient reasoning. As a result, our PSM completes large missing regions with photo-realistic content, as illustrated in Figure 1.

Our contributions can be summarized as follows:

- We develop a novel pixel spread model (PSM) customized for large-hole image inpainting. Thanks to the proposed iteratively decoupled probabilistic modeling, our model achieves efficient optimization and high-quality completion.
- Our method reaches cutting-edge performance on both Places Zhou et al. (2017) and CelebA-HQ Karras et al. (2018) benchmark datasets. Notably, our PSM outperforms popular denoising diffusion models, *e.g.*, LDM Rombach et al. (2022), by a large margin, yielding 1.1 FID improvement on Places2 Zhou et al. (2017) while being significantly more light-weighted (only 20% parameters, $10\times$ faster).

## 2 RELATED WORK

### 2.1 TRADITIONAL METHODS

Image inpainting is a classical computer vision problem. Early methods make use of image priors, such as self-similarity and sparsity. Diffusion-based methods Bertalmio et al. (2000); Ballester et al. (2001), for instance, convey information to the holes from nearby undamaged neighbors. Another line of exemplar-based approaches Hays & Efros (2007); Sun et al. (2005); Le Meur et al. (2011); Criminisi et al. (2003); Ding et al. (2018); Lee et al. (2016) looks for highly similar patches to complete missing regions using human-defined distance metrics. The most representative work is

PatchMatch Barnes et al. (2009), which employs heuristic searching in a multi-scale image space to speed up inpainting greatly. However, due to a lack of context understanding, they do not guarantee visually appealing and semantically consistent results.

## 2.2 Deep Learning Based Methods

Using a great amount of training data to considerably increase the ability of high-level understanding, deep-neural-network-based methods Pathak et al. (2016); Yan et al. (2018); Zeng et al. (2019); Liu et al. (2020); Wang et al. (2018b) achieve success. Pathak *et al*. Pathak et al. (2016) introduce the adversarial loss Goodfellow et al. (2014) to inpainting, yielding visually realistic results. Several approaches along this line continually push the performance to new heights. For example, in order to obtain locally fine-grained details and globally consistent structures, Iizuka *et al*. Iizuka et al. (2017) adopt two discriminators for adversarial training. Additionally, partial Liu et al. (2018) and gated Yu et al. (2019) convolution layers are proposed to reduce artifacts, *e.g.*, color discrepancy and blurriness, for irregular masks. Moreover, intermediate cues, including foreground contours Xiong et al. (2019), object structures Nazeri et al. (2019); Ren et al. (2019), and segmentation maps Song et al. (2018) are used in multi-stage generation. Despite nice inpainting content for small masks, these methods still do not guarantee large-hole inpainting quality.

## 2.3 Large Hole Image Inpainting

To deal with large missing regions, a surge of effort was made to improve the model capability. Attention techniques Yu et al. (2018); Liu et al. (2019); Xie et al. (2019); Yi et al. (2020) and transformer architectures Wan et al. (2021); Zheng et al. (2021); Li et al. (2022); Ko & Kim (2023) take advantage of context information. They work well when an image contains repeating patterns. Besides, Zhao *et al*. Zhao et al. (2020) propose a novel architecture, bridging the gap between image-conditional and unconditional generation, improving free-form large-scale image completion. There are also attempts to study the progressive generation. This line is to select only high-quality pixels each time and gradually fill holes. We note that these methods heavily rely on specially designed update algorithms Zhang et al. (2018a); Guo et al. (2019); Li et al. (2020); Oh et al. (2019), or consume additional model capacity to separately assess the prediction accuracy Zeng et al. (2020), or need more training stages Chang et al. (2022) when processing images.

Recently, benefiting from exact likelihood computation and iterative samplings, autoregressive models Wan et al. (2021); Yu et al. (2021); Wu et al. (2022) and denoising diffusion models Saharia et al. (2022a); Rombach et al. (2022); Lugmayr et al. (2022); Avrahami et al. (2022); Zhang et al. (2023) have shown great potential in producing diversified and realistic content. They inevitably incur high inference costs with thousands of steps and require massive computation resources. In this work, we present decoupled probabilistic modeling that obtains predictions and uncertainty measures simultaneously. Our model identifies reliable predicted pixels and sends them to subsequent iterations, thereby mitigating GANs-generated artifacts. Also, the proposed approach can be viewed as a diffusion model that learns pixel spreading rather than denoising and requires fewer iterations.

## 3 Our Method

Our objective is to use photo-realistic material to complete a masked image with substantial missing areas. In this section, we first formulate our pixel spread model (PSM) along with a comprehensive analysis. It is followed by the details of model design and loss functions.

## 3.1 Pixel Spread Model

Although GANs-based methods achieve significantly better results than traditional ones, they still face great difficulties handling large missing regions. We attribute one of the reasons to the one-shot nature of GANs and instead propose iterative inpainting.

In each pass, since there are inevitably some good predictions, we use these pixels as clues to assist the next-time generation. In this way, our pixel spread model gradually propagates valuable information to the entire image. In the following, we first discuss the single-pass modeling before moving on to the pixel spread process.

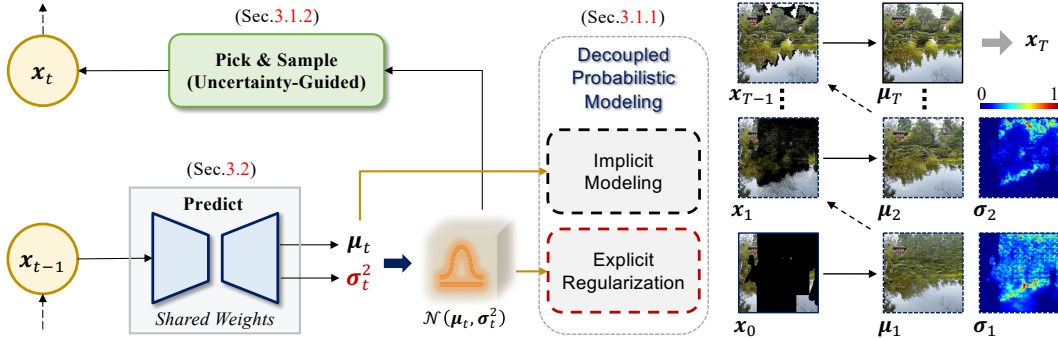

Figure 2: Our pixel spread model for high-quality large-hole image inpainting. Left illustration is the pixel spread pipeline with proposed decoupled probabilistic modeling, and the right images are visual examples. We simplify the input of the $t$-th iteration to $\boldsymbol{x}_{t-1}$, and denote the estimated mean and variance as $\boldsymbol{\mu}_t$ and $\boldsymbol{\sigma}_t^2$. The $\boldsymbol{\sigma}_t$ map on the right is normalized for better visualization. We observe gradual uncertainty reduction in missing regions during the pixel spread process.

### 3.1.1 DECOUPLED PROBABILISTIC MODELING

For iterative inpainting, it is essential to find a mechanism to evaluate the accuracy of predictions. One intuitive solution is introducing a tractable probabilistic model so that uncertainty information can be analytically calculated. However, this requirement often leads to the assumption that the approximated target distribution is Gaussian, which is considerably too simple to explain the truly complicated distributions. Although iterative models like denoising diffusion models Ho et al. (2020) enhance marginal distribution expression by including a number of hidden variables and optimizing the variational lower evidence bound, these methods typically yield a high inference cost.

To address these key issues, we propose a decoupled probabilistic modeling tailored for efficient iterative inpainting. The essential insight is that we leverage the advantages of implicit GANs-based optimization and explicit Gaussian regularization *in a decoupled way*. Thus we can simultaneously obtain accurate predictions and explicit uncertainty measures.

As shown in Figure 2, given an input image $\boldsymbol{x}_{t-1}$ at time $t$ with large holes, our model (see architecture details in Section 3.2) predicts the inpainting result $\boldsymbol{\mu}_t$ as well as an uncertainty map $\boldsymbol{\sigma}_t^2$. We use the adversarial loss (along with other losses of Section 3.3) to supervise image prediction $\boldsymbol{\mu}_t$, while jointly treating $(\boldsymbol{\mu}_t, \boldsymbol{\sigma}_t^2)$ as the mean and diagonal covariance of Gaussian distribution. GANs' implicit optimization makes it possible to approximate the true distribution as closely as possible, greatly reducing the number of iterations. It also supplies us with an explicit uncertainty measure for the mean term, allowing us to select reliable pixels. The Gaussian regularization is mainly applied to the variance term using negative log likelihood (NLL) $\mathcal{L}_{nll}$ as

$$\mathcal{L}_{nll} = -\sum_{i=1}^{D} \log \int_{\delta_-(\boldsymbol{y}^i)}^{\delta_+(\boldsymbol{y}^i)} \mathcal{N}\left(z; \text{sg}[\boldsymbol{\mu}_\theta^i(\boldsymbol{x})], \boldsymbol{\sigma}_\theta^i(\boldsymbol{x})^2\right) dz\,, \tag{1}$$

where $D$ is the data dimension and $i$ is the pixel index, $\theta$ denotes model parameters, input $\boldsymbol{x}$ and ground truth $\boldsymbol{y}$ are scaled to $[-1, 1]$, and $z$ follows the obtained Gaussian distribution $\mathcal{N}$. $\delta_+(y)$ and $\delta_-(y)$ are defined as

$$\delta_+(y) = \begin{cases} \infty & \text{if } y = 1\,, \\ y + \frac{1}{255} & \text{if } y < 1\,, \end{cases} \tag{2}$$

$$\delta_-(y) = \begin{cases} -\infty & \text{if } y = -1\,, \\ y - \frac{1}{255} & \text{if } y > -1\,. \end{cases} \tag{3}$$

Specifically, we include a stop-gradient operation (*i.e.*, sg[·]), which encourages the Gaussian constraint only to optimize the variance term and enables the mean term to be more accurately estimated through implicit modeling.

**Discussion.** We use the estimated mean and variance for sampling during the diffusion process, while taking the deterministic mean term as the output for the final iteration. The feasibility of

this design is proved by the experiments in Section 4. Additionally, the probabilistic modeling enables us to apply continuous sampling during pixel spread, yielding higher quality and more diverse estimations. Finally, we find the uncertainty measure also enables us to design a more effective attention mechanism in Section 3.2.

### 3.1.2 PIXEL SPREAD SCHEME

We use a feed-forward network, denoted as $f_\theta(\cdot)$, to gradually spread informative pixels to the entire image, starting from known regions as

$$\boldsymbol{x}_t, \boldsymbol{m}_t, \boldsymbol{u}_t = f_\theta(\boldsymbol{x}_{t-1}, \boldsymbol{m}_{t-1}, \boldsymbol{u}_{t-1}), \tag{4}$$

where $t$ is the time step, $\boldsymbol{x}_t$ refers to the masked image, $\boldsymbol{m}_t$ stands for a binary mask (1 for valid pixels while 0 for missing regions), and $\boldsymbol{u}_t$ is the uncertainty map. The output includes the updated image, mask, and uncertainty map. Network parameters are shared across all iterations.

We use several iterations for both training and testing to improve performance. Specifically, as shown in Figure 2 and Eq. (4), our method runs as follows at the $t$-th iteration.

1. **Predict.** Given the masked image $\boldsymbol{x}_{t-1}$, mask $\boldsymbol{m}_{t-1}$, and uncertainty map $\boldsymbol{u}_{t-1}$, our method estimates mean $\boldsymbol{\mu}_t$ and variance $\boldsymbol{\sigma}_t^2$ for all pixels. Then a preliminary uncertainty map $\tilde{\boldsymbol{u}}_t$ scaled to $[0,1]$ is generated by subtracting $\boldsymbol{\sigma}_t$'s min value and dividing by absolute max-min value. Note that the values of $\boldsymbol{\sigma}_t$ remain unchanged.

2. **Pick.** We first sort the uncertainty scores for missing regions based on $\boldsymbol{m}_{t-1}$. According to the pre-defined mask schedule, we calculate the number of pixels that will be added in this iteration, and insert those with the lowest uncertainty to the known category, updating the mask to $\boldsymbol{m}_t$. Based on the preliminary uncertainty map $\tilde{\boldsymbol{u}}_t$, by marking locations that are still missing as 1 and the initially known pixels as 0, while keeping $\tilde{\boldsymbol{u}}_t$ values of inpainted pixels up to this iteration (referring to $\boldsymbol{m}_t - \boldsymbol{m}_0$), we obtain the final uncertainty map $\boldsymbol{u}_t$.

3. **Sample.** We consider two situations. First, for the initially known locations based on $\boldsymbol{m}_0$, we always use the original input pixels $\boldsymbol{x}_0$ (0 for missing regions while other pixels are valid). Second, we apply continuous sampling in accordance with $\boldsymbol{\mu}_t$ and $\boldsymbol{\sigma}_t$ for the inpainting areas (referring to $\boldsymbol{m}_t - \boldsymbol{m}_0$). The result is formulated as

$$\boldsymbol{x}_t = \boldsymbol{x}_0 + (\boldsymbol{m}_t - \boldsymbol{m}_0) \odot (\boldsymbol{\mu_t} + \alpha \cdot \boldsymbol{\sigma}_t \odot \boldsymbol{z}), \tag{5}$$

where $\alpha$ is an adjustable ratio and $\boldsymbol{z} \sim \mathcal{N}(\boldsymbol{0}, \boldsymbol{I})$, and $\odot$ denotes Hadamard product. Note that the previously inpainted content gets updated at each iteration to maintain consistency with the newly inpainted pixels, and we do not use the $\boldsymbol{\sigma}_t \boldsymbol{z}$ term in the final iteration.

### 3.2 MODEL ARCHITECTURE

We use a deep U-Net Ronneberger et al. (2015) architecture with a StyleGAN Karras et al. (2019; 2020b) decoder, reaching large receptive fields with stacked convolutions to leverage context information in images Buades et al. (2005); Mairal et al. (2009); Berman et al. (2016); Wang et al. (2018a). In addition, we adopt multiple attention blocks at various resolutions, in light of the discovery that global interaction significantly improves reconstruction quality on much larger and more diverse datasets at higher resolutions Yu et al. (2018); Yi et al. (2020); Dhariwal & Nichol (2021).

Based only on feature similarity, the conventional attention mechanism Vaswani et al. (2017) offers equal opportunity for pixels to exchange information. For the inpainting task, however, missing pixels are initialized with the same specified values, making them close to one another. As a result, it is usually unable to effectively leverage useful information from visible regions. Even worse, the valid pixels are compromised, resulting in blurry content and unpleasing artifacts.

In this situation, as shown in Figure 3, we take into account the pixels' uncertainty scores to adjust the aggregating weights in attention by introducing a learnable function $\mathcal{F}$. It properly resolves the problem mentioned above. The attention output is computed by

$$\text{Attention}(\boldsymbol{q}, \boldsymbol{k}, \boldsymbol{v}, \boldsymbol{u}) = \text{Softmax}\left(\frac{\boldsymbol{q}\boldsymbol{k}^T}{\sqrt{d_k}} + \mathcal{F}(\boldsymbol{u})\right)\boldsymbol{v}, \tag{6}$$

where $\{\boldsymbol{q}, \boldsymbol{k}, \boldsymbol{v}\}$ are query, key, value matrices, $d_k$ denotes the scaling factor, and $\mathcal{F}$ predicts biased pixel weights using uncertainty map $\boldsymbol{u}$ and includes a reshape operation.

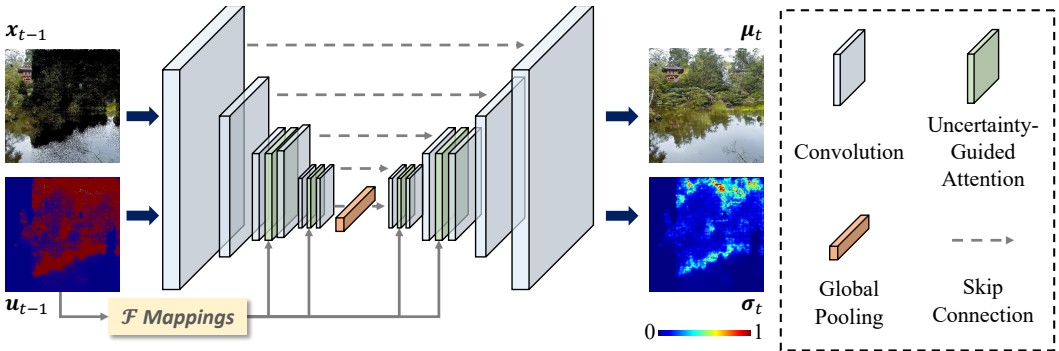

Figure 3: U-Net architecture with uncertainty-guided attention. We omit the mask update for clarity. The $\boldsymbol{\sigma}_t$ map is normalized for better visualization.

### 3.3 LOSS FUNCTIONS

In each iteration, our model outputs the mean and variance estimates, as shown in Figure 2. The mean term is optimized using adversarial loss Goodfellow et al. (2014) $\mathcal{L}_{adv}$ and perceptual loss Suvorov et al. (2021); Johnson et al. (2016) $\mathcal{L}_{pcp}$, which aims to produce natural-looking images. The losses are described as follows.

**Adversarial loss.** We formulate the adversarial loss as

$$\mathcal{L}_{ag} = -\mathbb{E}_{\hat{x}}\left[\log\left(D\left(\hat{x}\right)\right)\right] , \tag{7}$$

$$\mathcal{L}_{ad} = -\mathbb{E}_{x}\left[\log\left(D\left(x\right)\right)\right] - \mathbb{E}_{\hat{x}}\left[\log\left(1 - D\left(\hat{x}\right)\right)\right] , \tag{8}$$

where $D$ is the discriminator Karras et al. (2019), $x$ and $\hat{x}$ are real and predicted images.

**Perceptual loss.** We adopt a high receptive field perceptual loss Suvorov et al. (2021) as

$$\mathcal{L}_{pcp} = \sum_{i} \|\phi_i(x) - \phi_i(\hat{x})\|_2^2 , \tag{9}$$

where $\phi_i$ is the layer output of a pre-trained ResNet50 He et al. (2016).

As discussed in Section 3.1.1, we apply the negative log likelihood $\mathcal{L}_{nll}$ to constrain the variance for uncertainty modeling. Thus the final loss function for the generator is

$$\mathcal{L} = \sum_{j} \lambda_1 \mathcal{L}_{ag}^{j} + \lambda_2 \mathcal{L}_{pcp}^{j} + \lambda_3 \mathcal{L}_{nll}^{j}, \tag{10}$$

where $j$ is the number of spread iterations. We empirically set $\lambda_1 = 1$, $\lambda_2 = 2$ and $\lambda_3$ to $1 \times 10^{-4}$.

## 4 EXPERIMENTS

### 4.1 DATASETS AND METRICS

We train our models at $512 \times 512$ resolution on Places2 Zhou et al. (2017) and CelebA-HQ Karras et al. (2018) in order to adequately assess the proposed method. Places2 is a large-scale dataset with nearly 8 million training images in various scene categories. Additionally, 36,500 images make up the validation split. During training, images undergo random flipping, cropping, and padding, while testing images are centrally cropped to the $512 \times 512$ size. For CelebA-HQ, we employ 24,183 and 2,993 images, respectively, to train and test our models. Following Yu et al. (2019); Zhao et al. (2020); Suvorov et al. (2021); Li et al. (2022), we use on-the-fly generated masks during training, where the detailed setup is from MAT Li et al. (2022). We evaluate all models using identical masks provided by Li et al. (2022) for fair comparisons. Besides, for evaluating model robustness, we use the same model to inpaint both small and large masks.

Despite being adopted in early inpainting work, L1 distance, PSNR, and SSIM Wang et al. (2004) are found not strongly associated with human perception when assessing image quality Ledig et al. (2017); Sajjadi et al. (2017). In this work, in light of Zhao et al. (2020); Li et al. (2022), we use FID Heusel et al. (2017), P-IDS Zhao et al. (2020), and U-IDS Zhang et al. (2018b), which robustly measures the perceptual fidelity of inpainted images, as more suitable metrics.

Table 1: Quantitative ablation study. Model "A" is the full model. Models "B" and "C" use fewer training iterations. We remove the decoupled probabilistic modeling (DPM), continuous sampling (CS), and uncertainty-guided attention (UGA) in models "D", "E", and "F". Model "G" adopts attention at $16 \times 16$ size.

Table 2: Quantitative ablation study of the number of testing iterations. As the number of iterations increases, the FID↓ gets better and then saturates.

| Model | Train Iter. | DPM | CS | UGA | Att. Res. | FID↓ |
|-------|-------------|-----|-----|-----|-----------|------|
| A | 3 | ✓ | ✓ | ✓ | 32,16 | **2.36** |
| B | 1 | - | - | ✓ | 32,16 | 2.95 |
| C | 2 | ✓ | ✓ | ✓ | 32,16 | 2.55 |
| D | 3 | | ✓ | ✓ | 32,16 | 2.49 |
| E | 3 | ✓ | | ✓ | 32,16 | 2.45 |
| F | 3 | ✓ | ✓ | | 32,16 | 2.44 |
| G | 3 | ✓ | ✓ | ✓ | 16 | 2.64 |

| Test Iter. | Model A | Model B |
|-----------|---------|---------|
| 4 | 2.23 | 2.27 |
| 5 | 2.16 | 2.20 |
| 6 | 2.12 | 2.16 |
| 7 | 2.09 | 2.14 |
| 8 | 2.07 | 2.12 |
| 9 | **2.05** | 2.11 |
| 10 | **2.05** | 2.11 |

## 4.2 IMPLEMENTATION DETAILS

We use an encoder-decoder architecture. The encoder is made up of convolution blocks, while the decoder is adopted from StyleGAN2 Karras et al. (2020b). The encoder's channel size starts at 64 and doubles after each downsampling until the maximum of 512. The decoder has a symmetrical configuration. We adopt attention blocks at $32 \times 32$ and $16 \times 16$ resolutions. The uncertainty map is initialized as "1 - mask" at the first iteration. Given an $H \times W$ input, we first downsample the feature size to $\frac{H}{32} \times \frac{W}{32}$ before returning to $H \times W$. More details are provided in Appendix A.

We train our models for 20M images on Places2 and CelebA-HQ using 8 NVIDIA A100 GPUs. We utilize exponential moving average (EMA), adaptive discriminator augmentation (ADA), and weight modulation training strategies Karras et al. (2020a); Li et al. (2022). The batch size is 32, and the learning rate is $1 \times 10^{-3}$. We employ an Adam Kingma & Ba (2015) optimizer with $\beta_1 = 0$ and $\beta_2 = 0.99$. We empirically set $\alpha = 0.01$ in Eq. (5) based on experimental results. During training, our model undergoes the entire pipeline with two iterations to enhance efficiency. However, during testing, the model iterates four times to achieve improved restoration results.

The fact that previous work Zhao et al. (2020); Li et al. (2022) trains models on Places2 with 50M or more images – much more extensive data than ours – evidences the benefit of our method. Additional training can further improve our approach, and yet 20M images already deliver cutting-edge performance. Our model's generalization ability is demonstrated in Appendices B and C.

## 4.3 ABLATION STUDY

For quick evaluation, we train our models for 6M images at $256 \times 256$ resolution using Places365-Standard, a subset of Places2 Zhou et al. (2017). We start with model "A", which employs our full designs and adopts three iterations during training.

**Iterative number.** Our core idea is to employ iterative optimization to enhance the generation quality. We adjust the iteration number and maintain the same setup during training and testing. As illustrated in Table 1, models with one and two iterations, dubbed "B" and "C", yield 0.59 and 0.19 FID decreases compared to model "A". Also, as shown in Figure K.1, adopting more iterations is capable of producing more aesthetically pleasing content. The first and third cases exhibit obviously fewer artifacts, and the arch in the second example is successfully restored after three iterations.

It is noted that we can test the system with a different iteration number from the training stage. Using more iterations results in higher FID performance, as demonstrated in Table 2, yet at the expense of longer inference time. Thus, there is a trade-off between inference speed and generation quality. Additionally, when comparing models "A" and "B", it is clear that introducing more iterations in the training process is beneficial. But the number of iterations in the inference stage is more important.

**Decoupled probabilistic modeling.** To deliver accurate prediction while supporting the uncertainty measure for iterative inpainting, we propose decoupled probabilistic modeling. When putting all supervision on the sampled result, we observe the training diminishes the variance term (close to 0 for all pixels). It is because, unlike denoising diffusion models that precisely quantify the noise levels at each step, our GANs-based method no longer provides specific optimization targets for the mean and variance terms. The variance term is underestimated for trivial optimization in this case. It renders the picking process less effective.

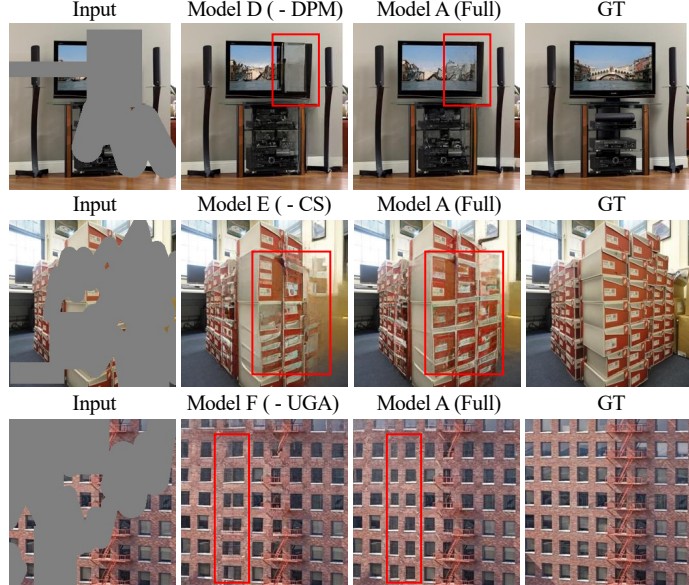

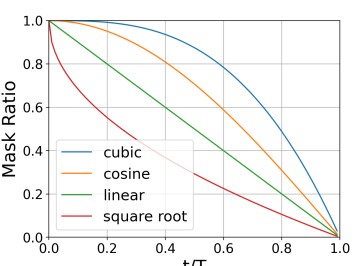

Figure 5: Visualization of mask schedule functions.

Figure 4: Qualitative ablation study. Model "A" is the full model. The proposed decoupled probabilistic modeling, continuous sampling, and uncertainty-guided attention designs are not used in models "D", "E", and "F".

Table 3: Ablation study of mask schedule functions.

| Mask Sche. | Iter. | FID↓ |
|---|---|---|
| Cubic | 3 | 2.54 |
| Cosine | 3 | 2.48 |
| Linear | 3 | **2.36** |
| Square Root | 3 | 2.47 |

As illustrated in Table 1, model "D" obtains an inferior FID result compared with the full model "A". Besides, from the visual comparison in Figure 4, it is observed that model "D" tends to generate blurry content, while model "A" produces sharper structures and fine-grained details.

**Continuous sampling.** Our approach uses the estimated variance to perform continuous sampling. Table 1 indicates that FID decreases by nearly 0.1 when continuous sampling (model "E") is not involved. Also, it is observed that our full model leads to more visually consistent content. For example, box structures are well restored from the visible pixels in Figure 4. Thus, continuous sampling brings higher fidelity to our results. As shown in Figure L.2, our model also supports the pluralistic generation, particularly in the hole's center. However, when the mean term is estimated with low uncertainty or the iteration number is constrained, the differences are not always instantly obvious. A detailed analysis of fidelity-diversity trade-off is further provided in Appendix H.

**Uncertainty-guided attention.** To fully exploit distant context, we add attention blocks to our framework. We first compare using attention at $32 \times 32$, $16 \times 16$ (model "A") and only at $16 \times 16$ (model "G"). We discover a 0.28 FID drop in model "G" from the quantitative comparison in Table 1, demonstrating the significance of long-range interaction in large-hole image inpainting.

Besides, as aforementioned in Section 3.2, the conventional attention mechanism may result in color consistency and blurriness. To support this claim, we tease apart the uncertainty guidance and notice a minor performance drop in Table 1. Also, we provide a visual comparison in Figure 4. We observe that model "A" produces more visually appealing window details than model "F".

**Mask schedule.** As illustrated in Table 3 and Figure 5, we analyze various mask schedule strategies and discover that the uniform strategy performs best. We argue this is because the mask ratios of input images vary widely, and uniform schedule results in more stable training for different iterations.

## 4.4    COMPARISONS TO STATE-OF-THE-ART METHODS

We thoroughly compare the proposed pixel spread model (PSM) with GANs-based models Li et al. (2022); Zhao et al. (2020); Suvorov et al. (2021); Zhu et al. (2021); Zeng et al. (2021); Yi et al. (2020); Yu et al. (2019), autoregressive models Wan et al. (2021), and denoising diffusion models Rombach et al. (2022) in Table 4. We use publicly accessible models for $512 \times 512$ resolution and test them on the same masks to make a fair comparison.

Table 4: Quantitative comparisons on Places2 and CelebA-HQ under $512 \times 512$ small and large mask settings. "†": Stable Diffusion inpainting model trained on LAION-Aesthetics V2 5+. P-IDS and U-IDS are shown as percentages(%). The best and second best results are in red and blue.

| Method | #Param. $\times 10^6$ | Places2 ($512 \times 512$) | | | | | | CelebA-HQ ($512 \times 512$) | | | | | |
| | | Small Mask | | | Large Mask | | | Small Mask | | | Large Mask | | |
| | | FID↓ | P-IDS↑ | U-IDS↑ | FID↓ | P-IDS↑ | U-IDS↑ | FID↓ | P-IDS↑ | U-IDS↑ | FID↓ | P-IDS↑ | U-IDS↑ |
| PSM (ours) | 74 | 0.72 | 30.95 | 43.91 | 1.68 | 25.33 | 39.30 | 2.34 | 22.42 | 33.43 | 4.05 | 16.10 | 28.25 |
| Stable Diffusion† | 860 | 1.32 | 12.69 | 34.78 | 2.11 | 12.01 | 32.57 | - | - | - | - | - | - |
| LDM | 387 | 1.06 | 16.23 | 39.61 | 2.76 | 12.11 | 33.02 | - | - | - | - | - | - |
| MAT | 62 | 1.07 | 27.42 | 41.93 | 2.90 | 19.03 | 35.36 | 2.86 | 21.15 | 32.56 | 4.86 | 13.83 | 25.33 |
| CoModGAN | 109 | 1.10 | 26.95 | 41.88 | 2.92 | 19.64 | 35.78 | 3.26 | 19.65 | 31.41 | 5.65 | 11.23 | 22.54 |
| LaMa | 51/27 | 0.99 | 22.79 | 40.58 | 2.97 | 13.09 | 32.29 | 4.05 | 9.72 | 21.57 | 8.15 | 2.07 | 7.58 |
| MADF | 85 | 2.24 | 14.85 | 35.03 | 7.53 | 6.00 | 23.78 | 3.39 | 12.06 | 24.61 | 6.83 | 3.41 | 11.26 |
| AOT GAN | 15 | 3.19 | 8.07 | 30.94 | 10.64 | 3.07 | 19.92 | 4.65 | 7.92 | 20.45 | 10.82 | 1.94 | 6.97 |
| HFill | 3 | 7.94 | 3.98 | 23.60 | 28.92 | 1.24 | 11.24 | - | - | - | - | - | - |

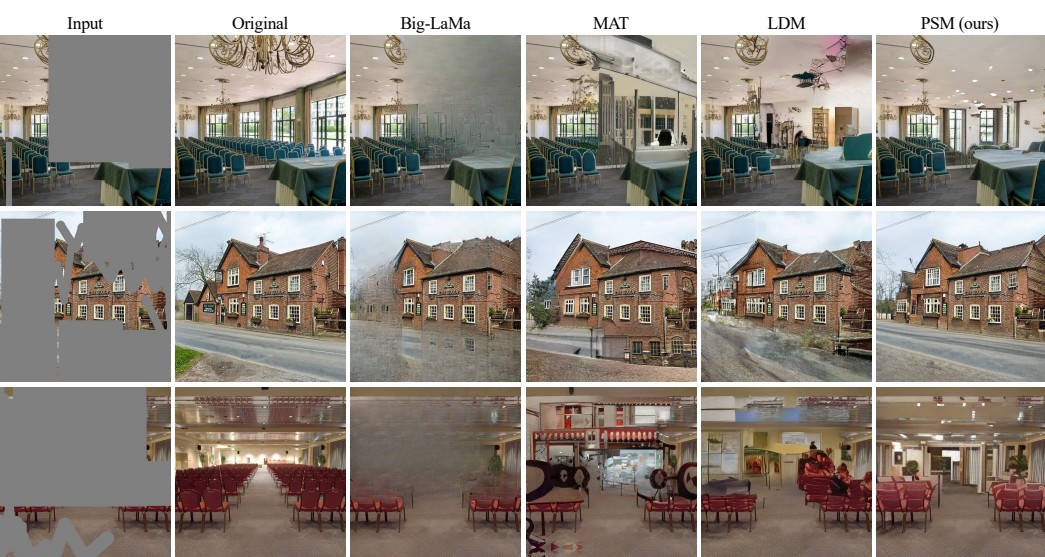

Figure 6: Qualitative comparisons of state-of-the-art methods on $512 \times 512$ Places2. Our PSM produces structures and details that are more realistic and reasonable. Best viewed zoomed in.

In Table 4, our method significantly performs better than the existing GANs-based models under both large and small mask settings. Besides, even with only 20% of the parameters of strong denoising diffusion model LDM Rombach et al. (2022), our method delivers superior results in terms of all metrics. For example, on the Places2 benchmark, our PSM brings about 1.1 improvement on FID and larger gains on P-IDS and U-ID under the large mask setup. As for the inference speed, our PSM costs nearly 250ms to obtain a $512 \times 512$ image, which is $10\times$ faster than LDM (~3s). Notably, our model is trained using far fewer samples (our 20M images vs. CoModGAN's Zhao et al. (2020) 50M images). Further comparisons are presented in Appendices D to G.

We also provide visual comparisons in Figure 6 and Appendix N. In a variety of scenes, our method generates more aesthetically pleasing textures with fewer artifacts when compared to existing methods. For instance, room layouts and building structures are better inpainted by our approach.

## 5 CONCLUSION

We have proposed a new pixel spread model for large-hole image inpainting. Utilizing the proposed iteratively decoupled probabilistic modeling, our method can assess the prediction accuracy and retain the pixels with the lowest uncertainty as hints for subsequent processing, yielding high-quality completion. Furthermore, our method exhibits favorable inference efficiency, largely surpassing that of prevalent denoising diffusion models. The state-of-the-art performance in multiple benchmarks demonstrates the effectiveness of our method. Lastly, we analyze limitations in Appendix M.

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

## A ARCHITECTURE DETAILS

Apart from the descriptions in Section 3.2 and Section 4.2, we here provide a more through illustration of architecture details. We adopt a U-Net architecture with skip connections, where the encoder downsamples the size of an $H \times W$ input to $\frac{H}{32} \times \frac{W}{32}$ and the decoder upsamples it back to $H \times W$. At each resolution, there is just one residual block made up of two $3 \times 3$ convolutional layers, unless otherwise stated. Both the encoder and the decoder employ attention blocks at feature sizes of $\frac{H}{16} \times \frac{W}{16}$ and $\frac{H}{32} \times \frac{W}{32}$, and an early convolutional block is also introduced at these scales. Different attention blocks use adaptive mapping functions in Figure 3, each of which is composed of 4 convolutional layers with a kernel size of $3 \times 3$.

The input consists of 7 channels: 3 for color images, 1 for the initial mask, 1 for the updated mask, 1 for the uncertainty map, and 1 for the time step. The number of channels is initially converted to 64, then doubled after each downsampling, up to a maximum of 512, and the decoder employs a symmetrical setup. The output contains 6 channels: 3 for the mean term, 3 for the log variance term.

We apply weight modulation, where the style is derived from an image global feature and a random latent code. As for the global feature, we employ convolutional layers to further downsample the feature size from $\frac{H}{32} \times \frac{W}{32}$ to $\frac{H}{256} \times \frac{W}{256}$ and a global pooling layer to obtain $1d$ representation. The random latent code is generated from Gaussian noise using 8 fully connected layers.

## B GENERALIZATION TO 1024×1024 RESOLUTION

To evaluate the generalization ability of models, we compare our pixel spread model (PSM), MAT Li et al. (2022) and LaMa Suvorov et al. (2021) trained on $512 \times 512$ Places2 Zhou et al. (2017) at the $1024 \times 1024$ resolution. As illustrated in Table B.1, our PSM performs significantly better than MAT and LaMa on all metrics, despite using fewer training samples. Remarkably, our approach results in an approximately 1.9 FID improvement. We do not involve denoising diffusion models (*e.g.*, LDM) and other GANs-based models (*e.g.*, CoModGAN) for comparisons because scaling them up to the $1024 \times 1024$ resolution is impractical.

Table B.1: Quantitative comparisons on $1024 \times 1024$ Places2 Zhou et al. (2017) dataset under the large mask setup by transferring models trained at the $512 \times 512$ resolution. Our PSM generalizes well to higher resolutions.

| Method | FID↓ | P-IDS(%)↑ | U-IDS(%)↑ |
|---|---|---|---|
| PSM (Ours) | **3.95** | **14.40** | **32.23** |
| MAT Li et al. (2022) | 5.83 | 9.51 | 28.02 |
| LaMa Suvorov et al. (2021) | 6.31 | 4.98 | 23.24 |

## C GENERALIZATION TO UNKNOWN MASK TYPES

Following the NTIRE 2022 Image Inpainting Challenge Romero et al. (2022) guidelines, we prepare a test set of 6000 samples with an average missing ratio of approximately $60\%$, covering mask types of Every N Lines, Image Expansion, and Nearest Neighbor, which are never seen during training.

Table C.2: Quantitative comparisons on unknown mask types.

| Method | #Param. | FID↓ | P-IDS↑ | LPIPS↓ | PSNR↑ |
|---|---|---|---|---|---|
| Ours + Fine-tune | 74M | **2.22** | **26.43** | **0.091** | **26.30**dB |
| Ours | 74M | **5.27** | **12.98** | **0.148** | **23.77**dB |
| Stable Diffusion | 860M | 95.65 | 3.65 | 0.705 | 13.56dB |
| LDM | 387M | 96.39 | 5.48 | 0.576 | 13.11dB |
| MAT | 62M | 67.57 | 3.28 | 0.458 | 16.25dB |
| Big LaMa | 51M | 47.64 | 3.97 | 0.306 | 20.41dB |

As shown in Table C.2, both other SOTA diffusion and GANs-based models show significant performance degradation, particularly for large models like Stable Diffusion and LDM. Surprisingly, our method exhibits excellent generalization on unknown mask types. **Our FID score of 5.27 far surpasses the second-best method, LaMa, with a score of 47.64.** We attribute this superiority to our iteratively decoupled probabilistic modeling, which enables the selection of reliable estimates for iterative refinement. Moreover, fine-tuning with a few iterations leads to great gains. These findings manifest the exceptional generalization and optimization abilities of our method.

## D   $512 \times 512$ LPIPS RESULTS

LPIPS Zhang et al. (2018b) is also a widely used perceptual metric in image inpainting. For a comprehensive comparison with state-of-the-art methods, we provide LPIPS results in Table D.3. We argue that LPIPS may not be suitable for large-hole image inpainting because it is calculated pixel-by-pixel. This measure is for reference only.

Table D.3: LPIPS↓ results on $512 \times 512$ Places2 Zhou et al. (2017) and CelebA-HQ Karras et al. (2018) datasets. "†": our models are trained with 20M samples, much less than other methods (*e.g.*, MAT uses 50M samples on Places2 and 25M samples on CelebA-HQ). We use a single model for both the small and large mask setups. "‡": the official Stable Diffusion inpainting model is trained on a large-scale high-quality dataset LAION-Aesthetics V2 5+.

| Method | #Param. $\times 10^6$ | Places | | CelebA-HQ | |
|---|---|---|---|---|---|
| | | Small | Large | Small | Large |
| PSM (Ours)† | 74 | **0.084** | **0.161** | **0.052** | **0.099** |
| Stable Diffusion‡ | 860 | 0.148 | 0.220 | - | - |
| LDM Rombach et al. (2022) | 387 | 0.100 | 0.190 | - | - |
| MAT Li et al. (2022) | 62 | 0.099 | 0.189 | 0.065 | 0.125 |
| CoModGAN Zhao et al. (2020) | 109 | 0.101 | 0.192 | 0.073 | 0.140 |
| LaMa Suvorov et al. (2021) | 51/27 | 0.086 | 0.166 | 0.075 | 0.143 |
| MADF Zhu et al. (2021) | 85 | 0.095 | 0.181 | 0.068 | 0.130 |
| AOT GAN Zeng et al. (2021) | 15 | 0.101 | 0.195 | 0.074 | 0.145 |
| HFill Yi et al. (2020) | 3 | 0.148 | 0.284 | - | - |

## E   $256 \times 256$ CELEBA-HQ RESULTS

We also conduct quantitative comparisons on $256 \times 256$ CelebA-HQ Karras et al. (2018) dataset. As shown in Table E.4, our method achieves the best performance among all methods.

Table E.4: Quantitative comparisons on $256 \times 256$ CelebA-HQ Karras et al. (2018) dataset. The P-IDS and U-IDS results are shown in percentage (%). "†": our model is trained with 12M samples, far less than other methods (*e.g.*, MAT uses 25M samples). We use a single model for both the small and large mask setups.

| Method | Small Mask | | | Large Mask | | |
|---|---|---|---|---|---|---|
| | FID↓ | P-IDS↑ | U-IDS↑ | FID↓ | P-IDS↑ | U-IDS↑ |
| PSM (Ours)† | **2.58** | **21.35** | **33.70** | **4.57** | **14.07** | **25.28** |
| MAT Li et al. (2022) | 2.94 | 20.88 | 32.01 | 5.16 | 13.90 | 25.13 |
| LaMa Suvorov et al. (2021) | 3.98 | 8.82 | 22.57 | 8.75 | 2.34 | 8.77 |
| ICT Wan et al. (2021) | 5.24 | 4.51 | 17.39 | 10.92 | 0.90 | 5.23 |
| RFR Li et al. (2020) | 6.37 | 5.75 | 14.97 | 12.91 | 0.70 | 1.77 |
| MADF Zhu et al. (2021) | 10.43 | 6.25 | 14.62 | 23.59 | 0.50 | 1.44 |
| AOT GAN Zeng et al. (2021) | 9.64 | 5.61 | 14.62 | 22.91 | 0.47 | 1.65 |
| DeepFill v2 Yu et al. (2019) | 5.69 | 6.62 | 16.82 | 13.23 | 0.84 | 2.62 |
| EdgeConnect Nazeri et al. (2019) | 5.24 | 5.61 | 15.65 | 12.16 | 0.84 | 2.31 |

## F  512×512 PSNR Results and Inference Speed

While PSNR may not be the most suitable metric for assessing large-scale hole inpainting performance, we provide the results in Table F.5 for reference. It is worth noting that our method demonstrates notably superior PSNR outcomes. In terms of inference efficiency, it is evident that our model stands out for its efficiency among the top-performing models.

Table F.5: PSNR (dB) results and inference speed on $512 \times 512$ Places2 Zhou et al. (2017) and CelebA-HQ Karras et al. (2018) datasets.

| Method | | PSM (Ours) | Stable Diffusion | LDM | MAT |
|---|---|---|---|---|---|
| Places | Small | **25.51**dB | 21.70dB | 24.48dB | 24.44dB |
| | Large | **20.89**dB | 19.17dB | 20.11dB | 19.92dB |
| CelebA-HQ | Small | **29.61**dB | - | - | 28.44dB |
| | Large | **24.81**dB | - | - | 23.50dB |
| Inference Speed | | **0.25**s | 3.6s | 2.7s | 0.26s |

## G  Comparison to RePaint

Considering that the sizes of RePaint Lugmayr et al. (2022) results are at $256 \times 256$ on Places2 and CelebA-HQ while ours are at $512 \times 512$, we don't compare it in the main body of the paper. Here we compare our model PSM to RePaint at $256 \times 256$ resolution on Places2 and CelebA-HQ in Table G.6, where PSM achieves better performance and is $1000\times$ faster than RePaint (*i.e.*, 0.25s v.s. 250s for one image processing). For saving time, we just use the first 10K Places2 validation images for evaluation.

Table G.6: Quantitative comparisons with RePaint Lugmayr et al. (2022) on $256 \times 256$ Places Zhou et al. (2017) and CelebA-HQ Karras et al. (2018) datasets.

| Method | Places2-10K ($256 \times 256$) | | | CelebA-HQ ($256 \times 256$) | | |
|---|---|---|---|---|---|---|
| | FID↓ | P-IDS(%)↑ | U-IDS(%)↑ | FID↓ | P-IDS(%)↑ | U-IDS(%)↑ |
| Ours | **3.47** | **18.32** | **34.52** | **4.57** | **14.07** | **25.28** |
| RePaint | 6.15 | 11.11 | 27.16 | 10.55 | 0.07 | 1.47 |

## H  Fidelity-Diversity Trade-Off

Apart from FID (depending on both diversity and fidelity), we follow previous work to use Improved Precision and Recall as fidelity (precision) and diversity (recall) measures. As shown in Table H.7, our model yields better FID, higher precision yet slightly lower recall than LDM on Places2, while outperforming MAT on all metrics. Improving diversity will be our future work.

Table H.7: FID, precision and recall comparisons for evaluating fidelity-diversity traed-off on $512 \times 512$ Places Zhou et al. (2017) dataset.

| Method | #Param. | FID↓ | Precision↑ | Recall↑ |
|---|---|---|---|---|
| PSM (Ours) | 74M | **1.68** | **0.983** | 0.971 |
| LDM | 387M | 2.76 | 0.962 | **0.975** |
| MAT | 62M | 2.90 | 0.965 | 0.939 |

# I  ADDITIONAL COMPARISONS

We have expanded our comparisons to include recent methods, including MI-GAN Sargsyan et al. (2023) and the inpainting model from ControlNet Zhang et al. (2023), as depicted in Table I.8. The results from our proposed PSM demonstrate significant improvements across all metrics, highlighting its effectiveness. MI-GAN is primarily designed for mobile devices, achieving a favorable performance-efficiency trade-off. Moreover, it is worth noting that ControlNet may produce suboptimal results due to its tendency to generate new objects that might not align harmoniously with the existing content.

Table I.8: Quantitative comparisons on Places2 and CelebA-HQ under $512 \times 512$ small and large mask settings. "†": Stable Diffusion inpainting model trained on LAION-Aesthetics V2 5+. P-IDS and U-IDS are shown as percentages(%). The best and second best results are in red and blue.

| Method | #Param. $\times 10^6$ | Places2 ($512 \times 512$) | | | | | | CelebA-HQ ($512 \times 512$) | | | | | |
| | | Small Mask | | | Large Mask | | | Small Mask | | | Large Mask | | |
| | | FID↓ | P-IDS↑ | U-IDS↑ | FID↓ | P-IDS↑ | U-IDS↑ | FID↓ | P-IDS↑ | U-IDS↑ | FID↓ | P-IDS↑ | U-IDS↑ |
| PSM (ours) | 74 | 0.72 | 30.95 | 43.91 | 1.68 | 25.33 | 39.30 | 2.34 | 22.42 | 33.43 | 4.05 | 16.10 | 28.25 |
| Stable Diffusion† | 860 | 1.32 | 12.69 | 34.78 | 2.11 | 12.01 | 32.57 | - | - | - | - | - | - |
| LDM | 387 | 1.06 | 16.23 | 39.61 | 2.76 | 12.11 | 33.02 | - | - | - | - | - | - |
| MAT | 62 | 1.07 | 27.42 | 41.93 | 2.90 | 19.03 | 35.36 | 2.86 | 21.15 | 32.56 | 4.86 | 13.83 | 25.33 |
| CoModGAN | 109 | 1.10 | 26.95 | 41.88 | 2.92 | 19.64 | 35.78 | 3.26 | 19.65 | 31.41 | 5.65 | 11.23 | 22.54 |
| LaMa | 51/27 | 0.99 | 22.79 | 40.58 | 2.97 | 13.09 | 32.29 | 4.05 | 9.72 | 21.57 | 8.15 | 2.07 | 7.58 |
| MI-GAN | 6 | 1.40 | 18.43 | 39.35 | 3.81 | 13.50 | 32.42 | - | - | - | - | - | - |
| ControlNet | 1223 | 1.86 | 12.63 | 35.71 | 5.55 | 6.60 | 25.65 | - | - | - | - | - | - |
| MADF | 85 | 2.24 | 14.85 | 35.03 | 7.53 | 6.00 | 23.78 | 3.39 | 12.06 | 24.61 | 6.83 | 3.41 | 11.26 |
| AOT GAN | 15 | 3.19 | 8.07 | 30.94 | 10.64 | 3.07 | 19.92 | 4.65 | 7.92 | 20.45 | 10.82 | 1.94 | 6.97 |
| HFill | 3 | 7.94 | 3.98 | 23.60 | 28.92 | 1.24 | 11.24 | - | - | - | - | - | - |

# J  RATIO $\alpha$ IN EQ. (5)

From Table J.9, we see that a large $\alpha$ generally trades fidelity (Precision) for higher diversity (Recall) on Places2. We empirically choose the $\alpha = 0.001$ for better evaluation results.

Table J.9: Quantitative results using different $\alpha$ values in Eq. (5).

| $\alpha$ Value | 0.001 (Ours) | 0.1 |
| --- | --- | --- |
| FID↓/Precision↑/Recall↑ | **1.68/0.983**/0.971 | 1.75/0.977/**0.975** |

# K  VISUAL ITERATION PROCESS

We depict the evolving results at various iterations in Figure K.1. It is evident that our method attains promising outcomes within a limited number of iterations, significantly faster than autoregressive and denoising diffusion models. This point is already underscored in the speed comparison discussed in Section 4.4 and Appendix F.

# L  PLURALISTIC GENERATION

As discussed in Section 4.3, our method also supports pluralistic generation. From the visual examples in Figure L.2, we observe that the differences mainly lie in the fine-grained details. We will work on improving the generating diversity.

# M  LIMITATION ANALYSIS

Our method shows a tendency to make more changes in small details rather than in large structures. We aim to improve the diversity of our generation in this regard. Additionally, our method some-

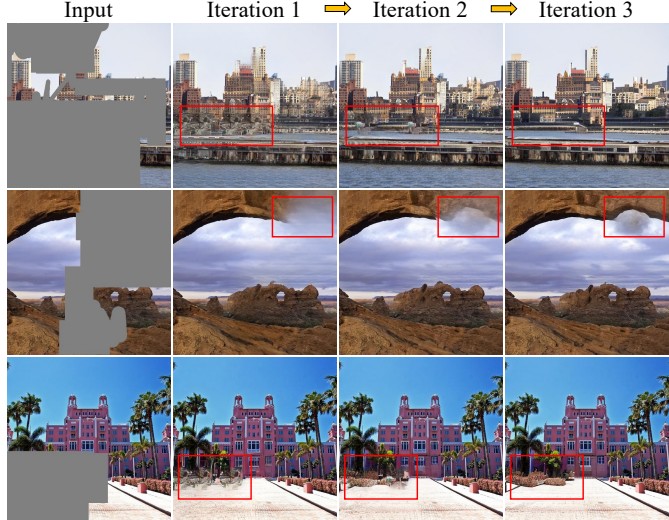

Figure K.1: Inpainting results of our PSM at different iterations. One-shot generation usually results in blurry content with unpleasing artifacts, while more iterations yield better results.

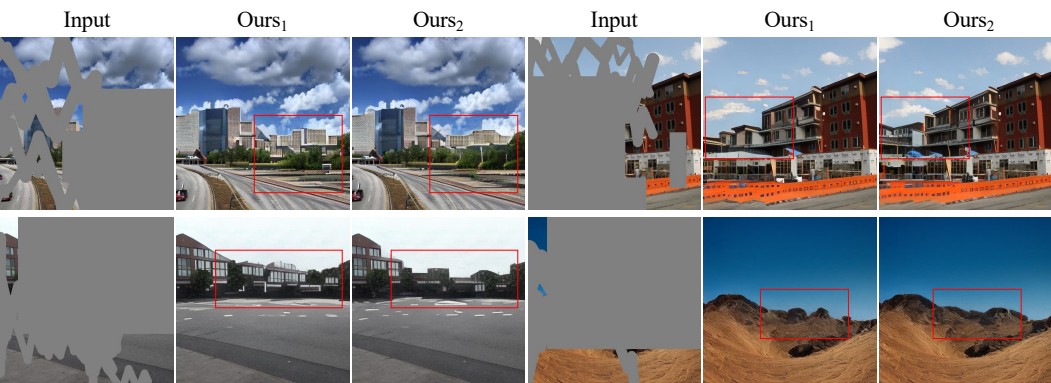

Figure L.2: Visual examples of diverse generation for our method.

times struggles to understand objects when only a few hints are given, as illustrated by a few failure cases presented in Figure M.3. For instance, the missing part of the notebook is filled with the background, and the recovered bus structure is incomplete. We attribute one of the reasons to the lack of high-level semantic understanding. We will further improve the generative capability of our model.

# N  ADDITIONAL QUALITATIVE COMPARISONS

We provide more visual examples on $512 \times 512$ Places2 Zhou et al. (2017) and CelebA-HQ Karras et al. (2018) in Figures N.4 to N.8. Due to space limit, we additionally add comparisons with CoModGAN Zhao et al. (2020) in Figure N.9. Compared to other methods, our method generates more photo-realistic and semantically consistent content. For example, our method successfully recovers human legs, airplane structures, and more realistic indoor and outdoor scenes.

Our model performs well when the input image contains sufficient visible pixel information, enabling high-quality generation while maintaining coherence with the existing content. This success originates from our model's pixel spreading mechanism, which initiates from visible pixels and progressively diffuses valuable information throughout the image. This approach is particularly effective for facial images characterized by strong inherent priors, such as symmetry and structural attributes. In the third example in Figure N.7, where the right half of the face is visible, our model

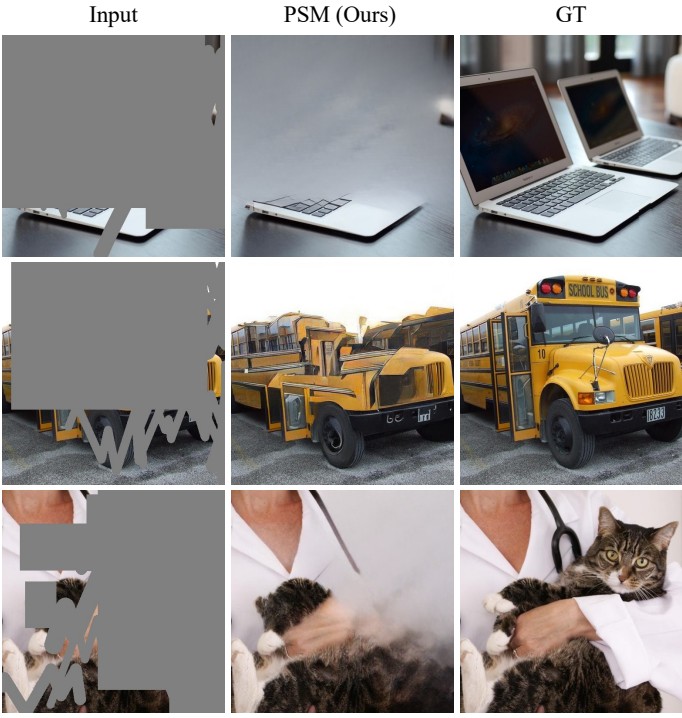

Figure M.3: Failure cases of our PSM. It is difficult to recover the large-scale missing objects.

reconstructs a comparatively realistic and consistent result using visible features like eyes and beard, albeit with potential discrepancies in details like ears and nose. However, when the entire face is masked, as in the first example in Figure N.7, our generated facial image notably diverges from ground truth. This phenomenon also appears in natural scenes; for instance, in the third example in Figure N.4, our method successfully restores the airplane structure due to the visibility of the frontal section, while other methods fail. All the results demonstrate the effectiveness of our method.

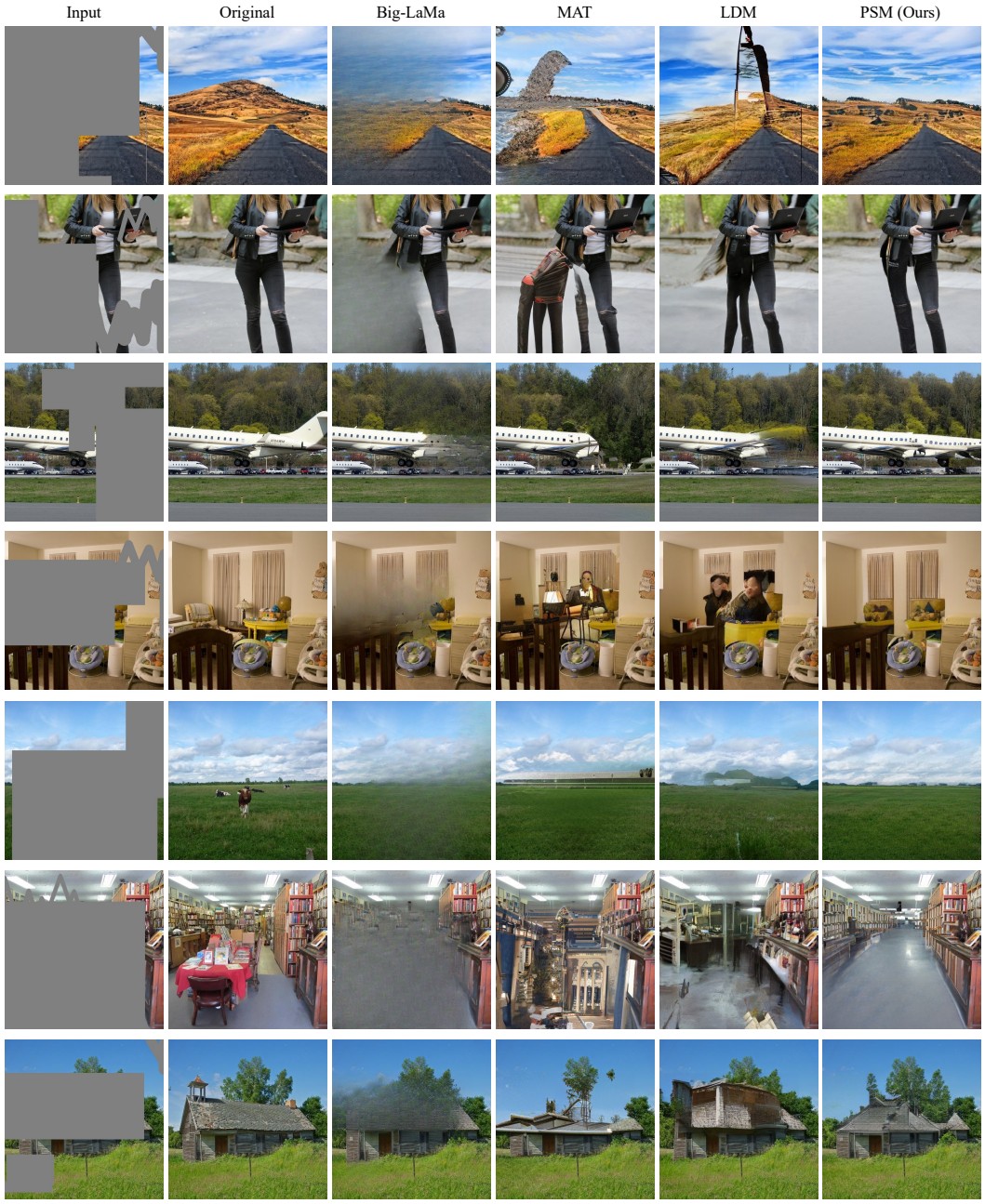

Figure N.4: Qualitative side-by-side comparisons of state-of-the-art methods on $512 \times 512$ Places2 dataset. Please zoom in for a better view. Our PSM produces structures and details that are more realistic and reasonable. Best viewed zoomed in.

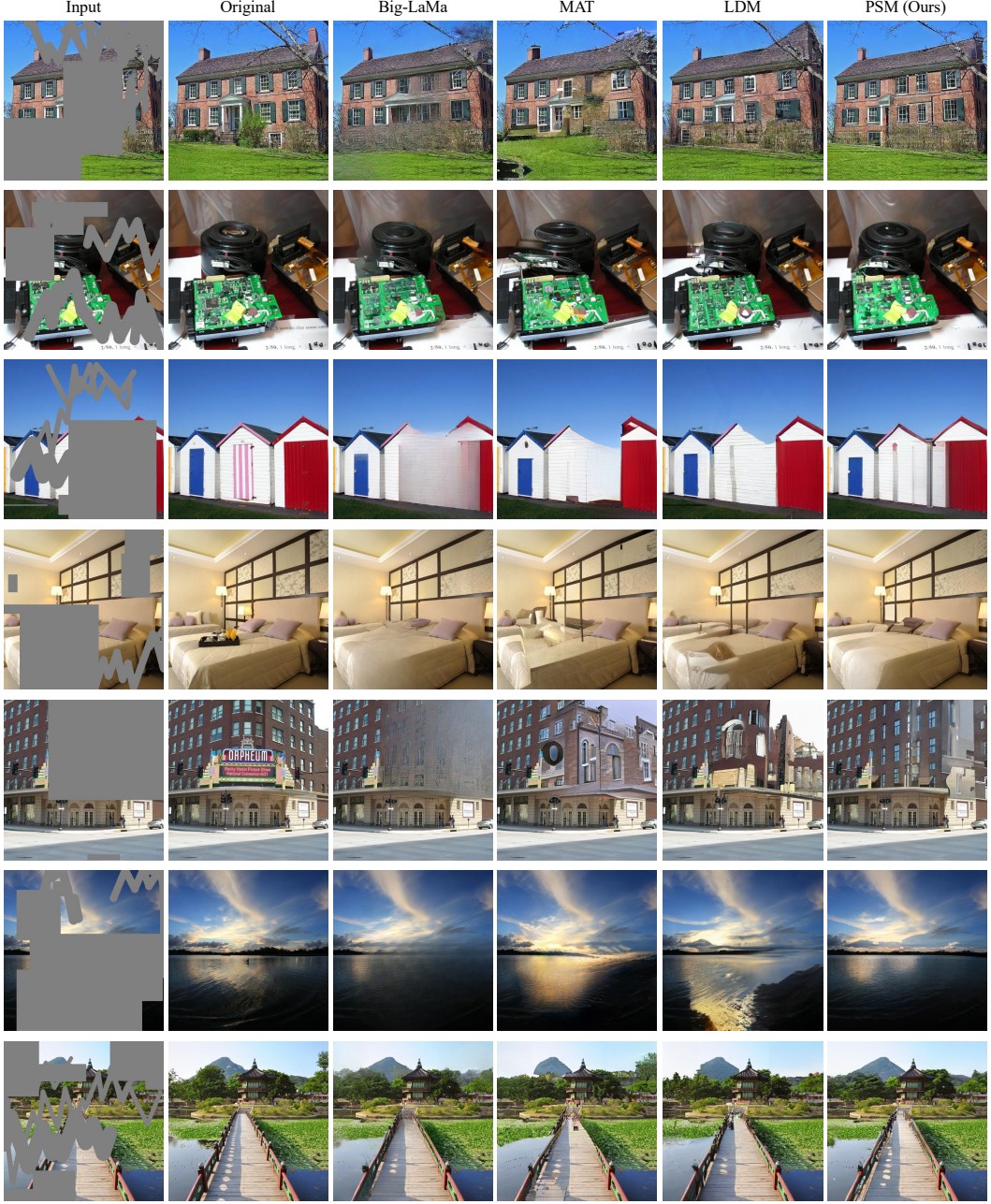

Figure N.5: Qualitative side-by-side comparisons of state-of-the-art methods on $512 \times 512$ Places2 dataset. Please zoom in for a better view. Our PSM produces structures and details that are more realistic and reasonable. Best viewed zoomed in.

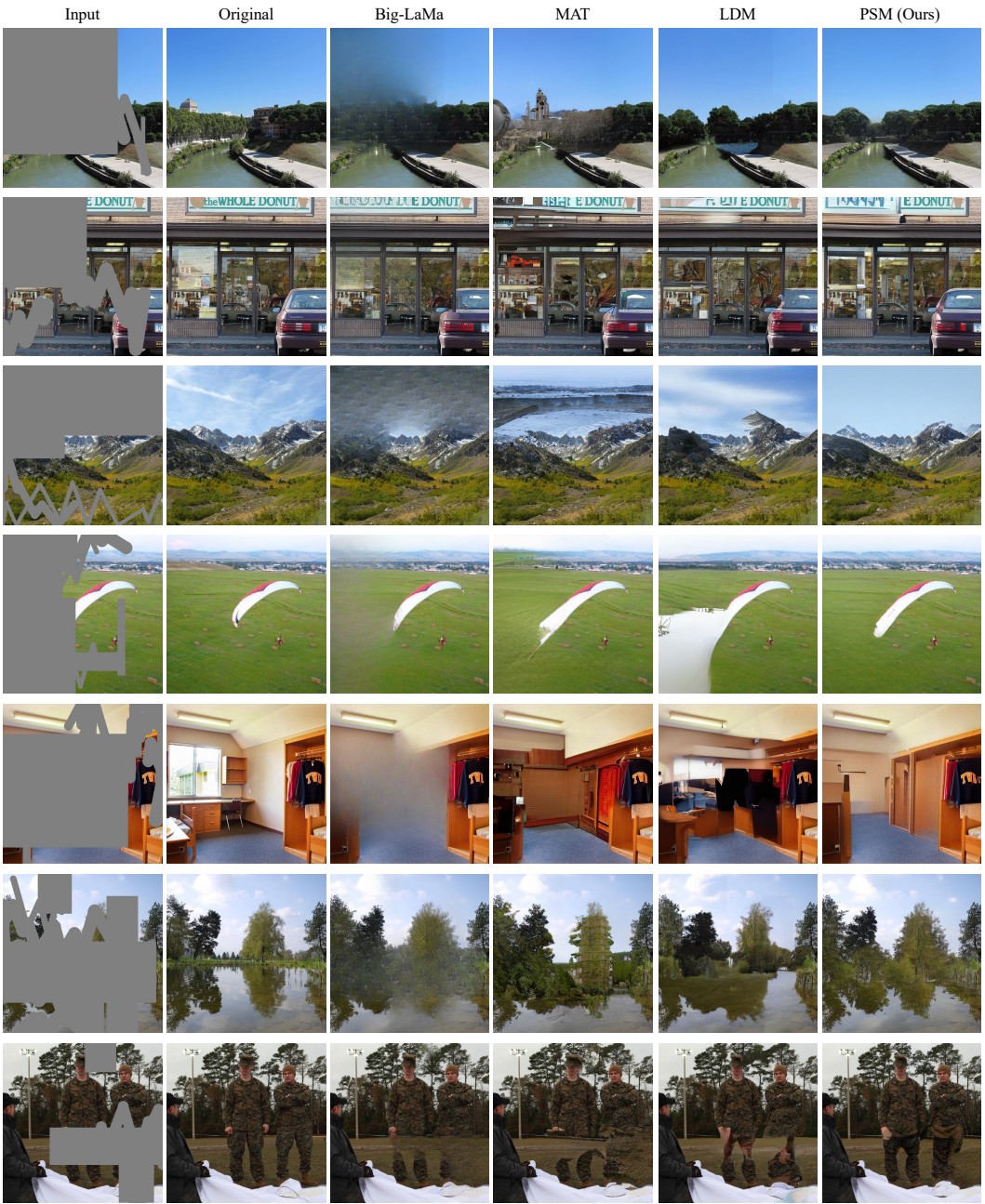

Figure N.6: Qualitative side-by-side comparisons of state-of-the-art methods on $512 \times 512$ Places2 dataset. Please zoom in for a better view. Our PSM produces structures and details that are more realistic and reasonable. Best viewed zoomed in.

| Input | Original | MADF | CoModGAN | MAT | PSM (Ours) |
|-------|----------|------|----------|-----|------------|

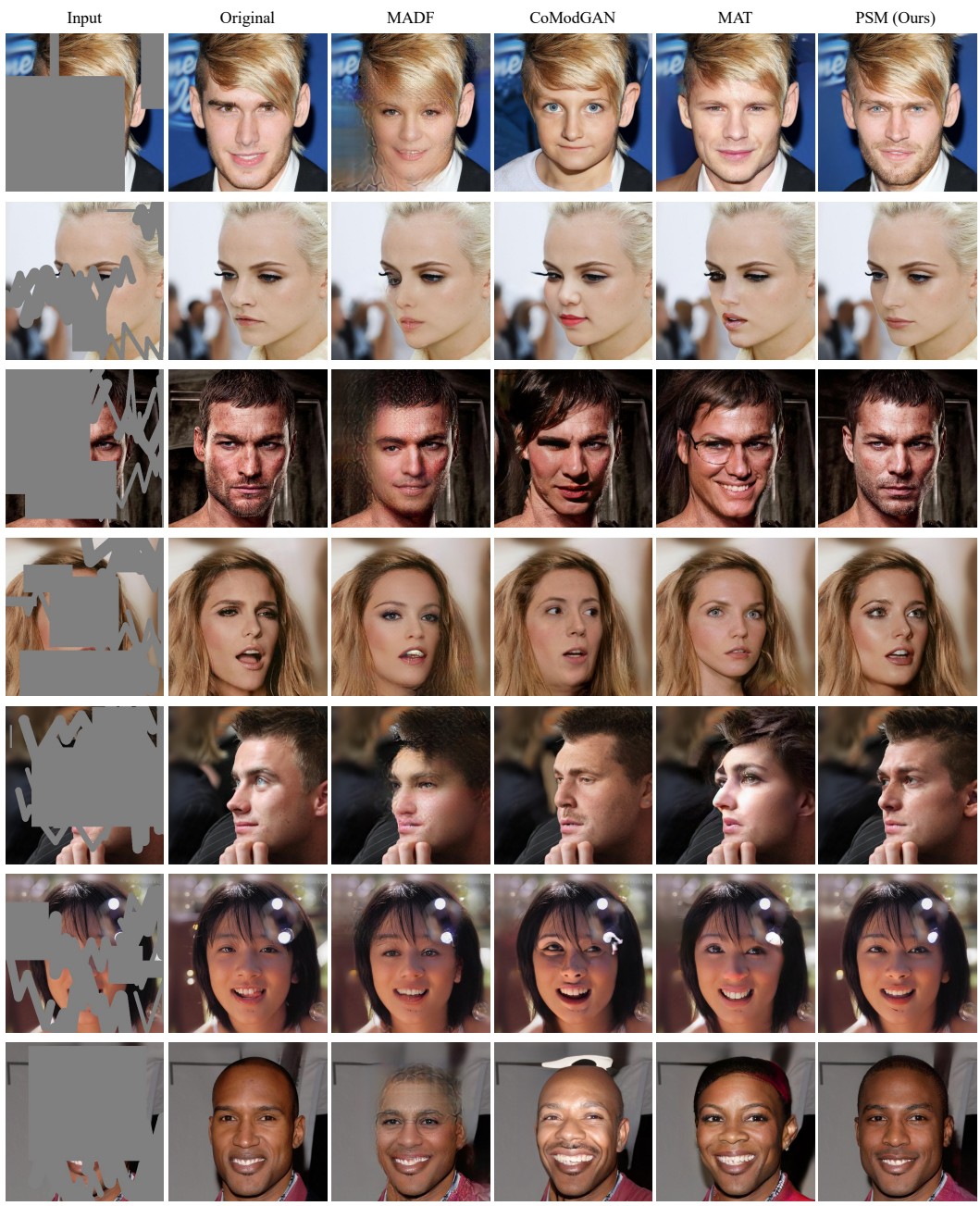

Figure N.7: Qualitative side-by-side comparisons of state-of-the-art methods on $512 \times 512$ CelebA-HQ dataset. Please zoom in for a better view. Our PSM produces face outlines and details that are more realistic and reasonable. Best viewed zoomed in.

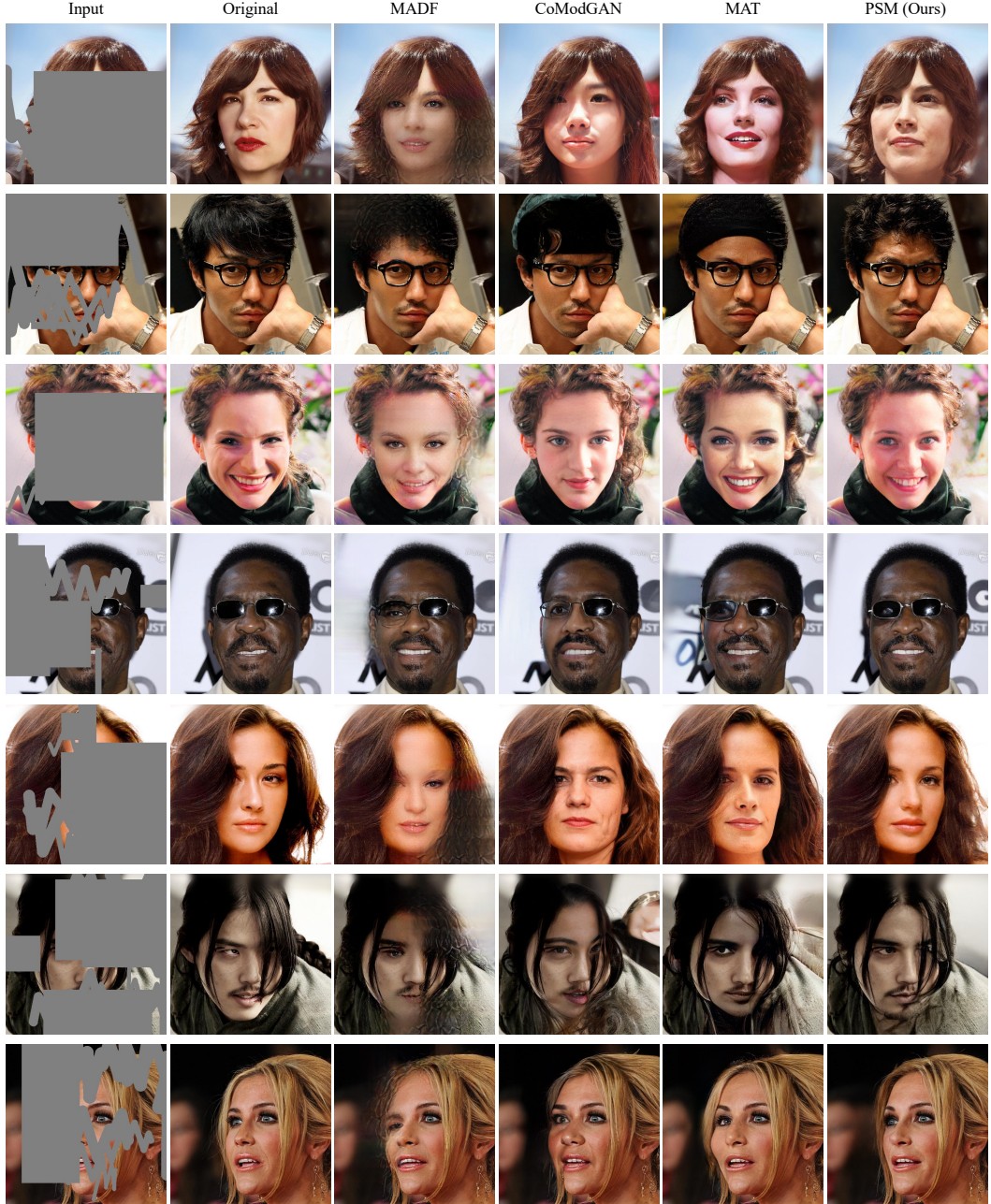

Figure N.8: Qualitative side-by-side comparisons of state-of-the-art methods on $512 \times 512$ CelebA-HQ dataset. Please zoom in for a better view. Our PSM produces face outlines and details that are more realistic and reasonable. Best viewed zoomed in.

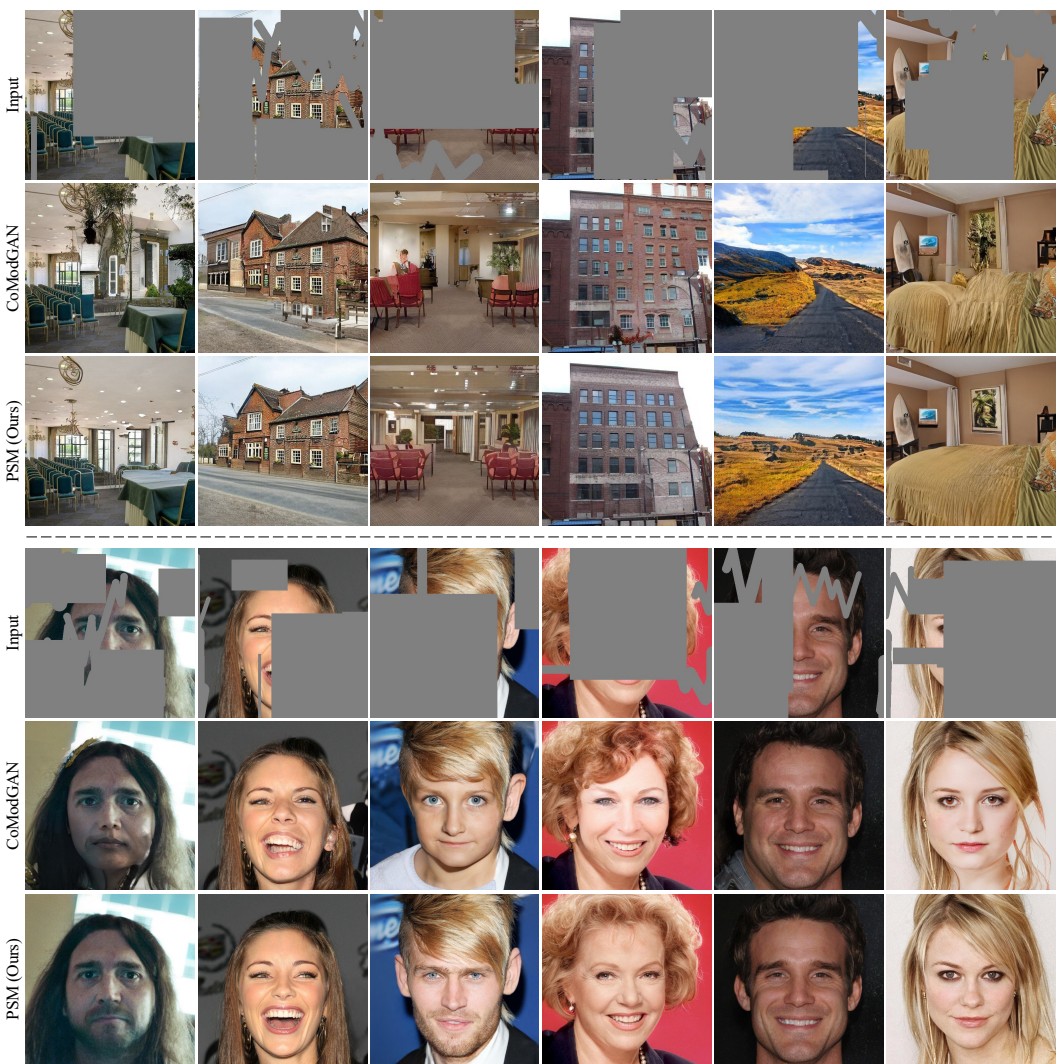

Figure N.9: Qualitative comparisons between CoModGAN and our PSM on $512 \times 512$ Places2 and CelebA-HQ datasets. Please zoom in for a better view. Our PSM produces structures and details that are more realistic and reasonable.

