# OpenReview forum: "Image Inpainting via Iteratively Decoupled Probabilistic Modeling"
_ICLR.cc/2024/Conference — ICLR 2024 spotlight_

### Official Review · Reviewer_rxeq · 2023-10-29

**Soundness:** 2 fair
**Presentation:** 3 good
**Contribution:** 2 fair
**Rating:** 6
**Confidence:** 3

**Summary:**

In this paper, the authors propose a novel method to tackle high-resolution image inpainting problems using an iteratively decoupled probabilistic modeling approach. In particular, given a masked image, the proposed method iteratively inpaints masked pixels by a GAN model, as well as the variance of the generated pixel. By keeping the low variance pixels the masked area is reduced. Then this step is repeated until the entire masked area is filled.

The authors did comprehensive evaluations with other methods and showed that the proposed method achieves the best performance and is more computationally efficient.

**Strengths:**

The originality of the proposed method mainly comes from the decoupling of mean and variance estimation for each pixel. In particular, at each iteration, the variance gives a metric to select the high-quality pixels as the input of the next iteration so that GAN artifacts are reduced. Also, since at each iteration, the entire image is generated, the whole process is scalable to high-resolution and large-hole inpainting problems. The experiments show strong numerical evidence of the performance of the method. Overall the paper is well-written and solid.

**Weaknesses:**

Overall the paper is well-written and sound. I have some specific questions regarding the math expressions (see the sections below).

**Questions:**

1. In Eq. 1, does $\mathcal(N)$ mean the density function of the Gaussian distribution?
2. In Eq. 1, what's the relationship between the $y$ being integrated and the $y$ in the integral bounds? Are they the same or different?
3. In Eq. 1, where does $y_i$ come from during training? Is it the ground-truth unmasked image or the GAN-generated one? If it is the latter, it also depends on $\theta$?
4. Above Eq. 9, there seems to be a typo: 'filed' -> 'field'

---

> ### Author Response · Authors · 2023-11-18
> **Response to Reviewer rxeq**
>
> Thanks for your comments. We have carefully revised our paper according to your suggestions.
>
> 1. **In Eq. 1, does $\mathcal N$ mean the density function of the Gaussian distribution?**
>
>     Correct, $\mathcal N(\cdot, \cdot)$ is the density function of Gaussian.
>
> 2. **In Eq. 1, what's the relationship between the $y$ being integrated and the $\boldsymbol y$ in the integral bounds? Are they the same or different?**
>
>     They are different. The $\boldsymbol y$ in the integral bounds refers to pixel values of the ground-truth unmasked image. Meanwhile, the $y$ being integrated follows a Gaussian distribution with the estimated mean and variance. We appreciate your highlighting this distinction. For improved clarity, we have represented the integrated variable as $z$ in the revised manuscript.
>
> 3. **In Eq. 1, where does $\boldsymbol y^i$ come from during training? Is it the ground-truth unmasked image or the GAN-generated one? If it is the latter, it also depends on $\boldsymbol \theta$?**
>
>     The $\boldsymbol y^i$ denotes the $i$-th pixel of the ground-truth unmasked image. We have explicitly clarified this in the revised manuscript.
>
> 4. **Above Eq. 9, there seems to be a typo: 'filed' -> 'field'.**
>
>     Thanks for pointing this out. We have fixed this typo.

---

> ### Author Response · Authors · 2023-11-23
> **Response to Reviewer rxeq**
>
> Dear Reviewer rxeq,
>
> Thank you for your detailed review and the valuable feedback. We have carefully provided clarifications and experiments to your previous comments. We appreciate your thorough evaluation of our work and look forward to hearing from you and addressing any further questions or concerns you may have.
>
> Thank you for your continued engagement and support.

---

### Official Review · Reviewer_N7bD · 2023-10-31

**Soundness:** 3 good
**Presentation:** 3 good
**Contribution:** 3 good
**Rating:** 8
**Confidence:** 4

**Summary:**

Authors present an iterative model for image in-painting, aiming at
filling large-holes with realistic looking areas. The model uses a GAN
and a Gaussian model to estimate an uncertainty term, which as far as
I can understand aims to find a covariance matrix that will minimize
negative log-likelihood of real intensities with respect to a Gaussian
model that uses the GAN output as the mean. Thus, this model uses the
covariance to account for the synthesis error. Pixels with low
variance are added to the current prediction and the process is
repeated until the entire empty region is filled. Experiments with
large-scale data sets are given.

**Strengths:**

1. A very simple method that gives really good results.
2. Authors experimented with large-scale data sets.
3. Results are very impressive. The presented comparisons with
   state-of-the-art methods shows that the proposed iterative scheme
   yields better in-painting than all the rest. Furthermore, it does
   it in a fraction of the time compared to the most recent methods. I
   believe these results are extremely promising, and if they can be
   reproduced they may have a substantial impact.

**Weaknesses:**

1. The training strategy is not well explained. Authors should clarify
   whether and how the iterations are taken into account during the
   training.
2. There are some suspiciously accurate in-painting results in the
   appendix - particularly in Fig. M7. Can authors explain how the
   proposed model can be so accurate? I can understand being
   realistic, however, the third column shows that the generation is
   pretty much the same. Similar in-apinting results are given in
   Figure M8.

**Questions:**

1. The in-painting results are suspiciously good in some
   cases. Can authors explain how their model can be so accurate - not
   just realistic - in M7 and M8? May there be a problem with training
   and test data leakage?

I am happy to increase my score, however, first I would like to wait authors' response to this question.

---

> ### Author Response · Authors · 2023-11-18
> **Response to Reviewer N7bD**
>
> 1. **The training strategy is not well explained. Authors should clarify whether and how the iterations are taken into account during the training.**
>
>     Thank you for your valuable advice. During training, our model undergoes the full pipeline with all iterations (we iterate twice for training efficiency, as stated in Sec.4.2), and the updating scheme between iterations is detailed in Sec.3.1.2. However, for testing, we may iterate for different numbers. As shown in Table 2, using more iterations results in higher FID performance. We have revised our manuscript to provide clearer illustrations of this point.
>
> 2. **There are some suspiciously accurate in-painting results in the appendix - particularly in Fig.N7 (Fig.M7 in the original paper corresponds to Fig.N7 in the revised version). Can authors explain how the proposed model can be so accurate? I can understand being realistic, however, the third column shows that the generation is pretty much the same. Similar inpainting results are given in Fig.N8. May there be a problem with training and test data leakage?**
>
>     There is no problem with training and test data leakage. Following established methodologies like LaMa [a] and MAT[b], we utilize the official training (24183 images) and validation (2993 images) sets of CelebA-HQ. We have carefully checked that individuals present in the validation set do not overlap with the training data.
>
>     Our model performs well when the input image contains sufficient visible pixel information, enabling high-quality generation while maintaining coherence with the existing content. This success originates from our model's pixel spreading mechanism, which initiates from visible pixels and progressively diffuses valuable information throughout the image.  This approach is particularly effective for facial images characterized by strong inherent priors, such as symmetry and structural attributes. In the third example in Figure N.7, where the right half of the face is visible, our model reconstructs a comparatively realistic and consistent outcome using visible features like eyes and beard, albeit with potential discrepancies in details like ears and nose. However, when the entire face is masked, as in the first example in Figure N.7, our generated facial image notably diverges from ground truth. This phenomenon also appears in natural scenes; for instance, in the third example in Figure N.4, our method successfully restores the airplane structure due to the visibility of the frontal part, while other methods fail to inpaint such regions.
>
>     We will release the training code and models. All results are reproducible.
>
>     [a] Suvorov, Roman, et al. "Resolution-robust large mask inpainting with fourier convolutions." WACV. 2022.
>     [b] Li, Wenbo, et al. "Mat: Mask-aware transformer for large hole image inpainting." CVPR. 2022.

---

> > ### Comment · Reviewer_N7bD · 2023-11-22
> > **thanks**
> >
> > Thanks for the clarifications. The quality of the results is quite impressive.

---

### Official Review · Reviewer_EPEv · 2023-10-31

**Soundness:** 3 good
**Presentation:** 3 good
**Contribution:** 3 good
**Rating:** 6
**Confidence:** 3

**Summary:**

In this paper, the authors propose the pixel spread model (PSM) to gradually fill the masked regions of an image. Specifically, the model learns the mean by GAN loss and the variance by negative log likelihood (NLL). In each iteration, the model first makes the prediction, then picks the most valuable pixels to fill the region and leave the others masked. Experiments show that the method works well.

**Strengths:**

- The idea is novel to gradually fill the regions through the computed mean and variance.
- The method is efficient by utilizing the optimized computational resource of GAN model and the thoughts of autoregressive model.

**Weaknesses:**

- Some parts of the paper is not very clear. For example,
    - What's the meaning of $y^i$ in equation (1)? Is it the groundtruth?
    - More explanation is necessary for equation (5). What happens when $m_t=1$ and $m_0=0$ or vise versa?
    - Is the uncertainty map computed only in the mask region or the entire image?
    - Why $\sigma_t$ is in the range $[0,1]$?

- Comparison to Controlnet is necessary.
- It seems that sometimes the model is hard to grasp some structure of the image. For example, in the last row of Fig.4, the result of Model A cannot reconstruct the structure of stairs.

**Questions:**

Please see above.

---

> ### Author Response · Authors · 2023-11-18
> **Response to Reviewer EPEv**
>
> 1. **What's the meaning of in Eq.1? Is it the ground truth?**
>
>     The $y^i$ represents the $i$-th pixel of the ground-truth image. We have revised the paper to improve the clarity of this illustration.
>
> 2. **More explanation is necessary for Eq.5. What happens when $\boldsymbol m_t=1$ and $\boldsymbol m_0=0$ or vice versa?**
>
>     In the mask $\boldsymbol m$, we denote known regions with values of 1 and missing regions as 0. In Eq.5, we assign pixel values in the missing regions to 0 while preserving the original pixel values in the initially known area. Subsequently, we progressively fill in the missing regions using sampled results based on newly estimated mean and variance values. It's important to note that the previously inpainted content in the missing regions is updated at each iteration to maintain consistency with the newly inpainted pixels.
>
>     In cases when $\boldsymbol m_0=0$, the initially visible pixels are absent. We consider this as a generation task. However, in our inpainting mask, such a phenomenon does not occur since we do not set $\boldsymbol m_0=0$. Furthermore, we only set $\boldsymbol m_t=1$ during the final iteration to ensure training convergence. When $\boldsymbol m_0=1$, all pixels are regarded as valid. So we will not update anymore.
>
> 3. **Is the uncertainty map computed only in the mask region or the entire image?**
>
>     The uncertainty map is computed for the entire image. As discussed in the second step (i.e., 'Pick'), the uncertainty score of each initially known pixel is set to 0. Additionally, for the pixels that have been inpainted up to the current time step, their uncertainty scores  are computed by using the $\boldsymbol \sigma_t$ map. For the pixels located in regions that are still missing, their values are set as 1.

---

> ### Author Response · Authors · 2023-11-18
> **Response to Reviewer EPEv (Part 2)**
>
> 4. **Why is $\boldsymbol \sigma_t$ in the range [0,1]?**
>
>     We do not constrain the range of $\boldsymbol \sigma_t$ to [0,1]. Instead, we normalize $\boldsymbol \sigma_t$ to compute the uncertainty map $\boldsymbol u_t$, which ranges from [0,1]. In the last sampling step, we use the original $\boldsymbol \sigma_t$ values. The $\boldsymbol \sigma_t$ maps in Figure 2 and Figure 3 are normalized only for better visualization. We have clarified this point in the revised paper.
>
> 5. **Comparison to Controlnet is necessary.**
>
>     Thank you for the constructive comments. In addition to ControlNet [a], we also include MI-GAN \[b\] (ICCV 2023) in our comparisons. Although ControlNet is not trained on the Places2 dataset, it undergoes training on a more extensive LAION dataset. Results from our proposed PSM show significant improvements across all metrics, highlighting the effectiveness of our method. We observe that ControlNet might produce undesired results due to its tendency to generate new objects that might not align harmoniously with the existing content. We have added the comparisons to Sec.I of the supplementary material due to the page limit.
>
>     #### 512x512 Places2 under the small mask ratio setting
>     | | | | | | |
>     |---|---|---|---|---|---|
>     | Method | FID$\downarrow$ | P-IDS$\uparrow$ | U-IDS$\uparrow$ | LPIPS$\downarrow$ | PSNR$\uparrow$ |
>     | **PSM (ours)** | **0.72** | **30.95** | **43.91** | **0.084** | **25.51** |
>     | MIGAN | 1.40 | 18.43 | 39.35 | 0.103 | 24.38 |
>     | ControlNet | 1.86 | 12.63 | 35.71 | 0.117 | 24.68 |
>     | | | | | | |
>
>     #### 512x512 Places2 under the large mask ratio setting
>     | | | | | | |
>     |---|---|---|---|---|---|
>     | Method | FID$\downarrow$ | P-IDS$\uparrow$ | U-IDS$\uparrow$ | LPIPS$\downarrow$ | PSNR$\uparrow$ |
>     |**PSM (ours)**| **1.68** | **25.33** | **39.30** | **0.161** | **20.89** |
>     |MIGAN| 3.81|13.50|32.42|0.195|20.00|
>     |ControlNet|5.55|6.60|25.65|0.219|19.73|
>     | | | | | | |
>
>
>     [a] Zhang, Lvmin, Anyi Rao, and Maneesh Agrawala. "Adding conditional control to text-to-image diffusion models." ICCV. 2023.
>     [b] Sargsyan, Andranik, et al. "MI-GAN: A Simple Baseline for Image Inpainting on Mobile Devices." ICCV. 2023.
>
> 6. **It seems that sometimes the model is hard to grasp some structure of the image. For example, in the last row of Fig.4, the result of Model A cannot reconstruct the structure of stairs.**
>
>     We agree that our model sometimes hard to grasp some structures during generation. This is attributed to our pixel spread model, which functions by spreading informative pixels from initially visible regions across the entire image. In the last example depicted in Figure 4, despite some remaining stairs, the spreading process appears to be primarily influenced by the window representation due to its frequent appearance in the input. Consequently, our model effectively reconstructs the windows. However, even in the absence of stairs, our output remains reasonable.
>
>     Indeed, it is very challenging to consistently generate all subtle structures with high fidelity, as the content of the free-form masked regions is typically complex and ambiguous. This challenging factor also deteriorates diffusion models during generation although they have shown great generation abilities, as shown in Figure 6. We will concentrate on improving the generation quality in our future work.

---

> ### Author Response · Authors · 2023-11-23
> **Response to Reviewer EPEv**
>
> Dear Reviewer EPEv,
>
> Thank you for your detailed review and the valuable feedback. We have carefully provided clarifications and experiments to your previous comments.  We appreciate your thorough evaluation of our work and look forward to hearing from you and addressing any further questions or concerns you may have.
>
> Thank you for your continued engagement and support.

---

### Official Review · Reviewer_HZRy · 2023-11-02

**Soundness:** 3 good
**Presentation:** 2 fair
**Contribution:** 3 good
**Rating:** 5
**Confidence:** 4

**Summary:**

This paper proposes a pixel spread model for image inpainting, which iteratively employs decoupled probabilistic modeling. Specifically, this work combines the optimization efficiency of GANs with the prediction tractability of probabilistic models and selectively spreads informative pixels throughout the image in a few iterations, improving the completion quality and efficiency. Experimental results show the proposed method achieves a state-of-the-art performance on several benchmarks.

**Strengths:**

The proposed pixel spread model makes use of the merits of GANs’ efficient optimization and the tractability of probabilistic models.

Good performance is achieved on several benchmark datasets.

**Weaknesses:**

There are several unclear statements listed as follows.

**Questions:**

1. Please explain in detail how the uncertainty is obtained in each iteration.
2. The third paragraph in Section 3.1.1 is inconsistent with Figure 2. The input image x_t at time t should be x_{t-1} at time t-1, right?
3. Please discuss in detail the effect of the learnable function F in Eq. (6).
4. The authors state several times that the proposed method improve the efficiency. Please demonstrate in detail, e.g., the comparison of inference time.
5. What is the mask size in Table 4?
6. In the experiments, please add the comparisons with more recent methods, e.g.,
[a] Chu et al., Rethinking Fast Fourier Convolution in Image Inpainting, ICCV 2023.
[b] Sargsyan et al., MI-GAN: A Simple Baseline for Image Inpainting on Mobile Devices, ICCV 2023.
[c] Ko et al., Continuously Masked Transformer for Image Inpainting, ICCV 2023.

---

> ### Author Response · Authors · 2023-11-18
> **Response to Reviewer HZRy**
>
> 1. **Please explain in detail how the uncertainty is obtained in each iteration.**
>
>     We provide the details of obtaining the uncertainty in each iteration. Taking the $t$-th iteration as an example, we forward the masked image $\boldsymbol x_{t-1}$, binary mask $\boldsymbol m_{t-1}$ and uncertainty map $\boldsymbol u_{t-1}$ to the network to obtain the estimated mean $\boldsymbol \mu_t$ and variance $\boldsymbol \sigma_t^2$. To obtain a preliminary uncertainty map $\tilde {\boldsymbol u}_t$ in \[0,1\], we normalize the standard deviation map $\boldsymbol \sigma_t$ by subtracting its min value and dividing by absolute max-min value (the smaller $\boldsymbol \sigma_t$, the lower uncertainty). Note that the final sampling (Step 3 in Sec.3.1.2) is performed using the original $\boldsymbol \sigma_t$ without normalization.
>
>     We then sort the uncertainty scores only for unknown pixels based on $\boldsymbol m_{t-1}$. According to the pre-defined mask schedule, we figure out the pixels that will be newly added in this iteration. Based on the preliminary uncertainty map $\tilde {\boldsymbol u}_t$, by marking pixel locations that are still missing as 1 and the initially known pixel locations (referring to $\boldsymbol m_0$) as 0, while keeping the $\tilde {\boldsymbol u}_t$ values of inpainted pixels unchanged, we obtain the final uncertainty map $\boldsymbol u_t$.
>
> 2. **The third paragraph in Section 3.1.1 is inconsistent with Figure 2. The input image $\boldsymbol x_t$ at time t should be $\boldsymbol  x_{t-1}$ at time $t-1$, right?**
>
>     Thanks for pointing this out. We have fixed this typo. It should be $\boldsymbol x_{t-1}$.
>
> 3. **Please discuss in detail the effect of the learnable function $\mathcal F$ in Eq.6.**
>
>     The learnable function $\mathcal F$ introduces the proposed uncertainty-guided attention mechanism. In traditional attention based solely on feature similarity, all pixels have an equal chance to exchange information. However, in image inpainting, where missing pixels are initialized with specified identical values, conventional attention mechanisms are not effective in conveying valuable information from visible regions to missing regions. This phenomenon often yields unsatisfied pixels, resulting in blurry content and undesirable artifacts.
>
>     By incorporating the uncertainty map as an input, the function $\mathcal F$ discerns between valid and invalid pixels. This enables our network to prioritize attention on valid pixels when filling in the holes. Removal of this design (i.e., 'model F' in Table 1) results in a performance drop compared to our full 'model A'. Additionally, the structures are sharper and clearer in the inpainted results via our uncertainty-guided attention in Figure 4.

---

> > ### Author Response · Authors · 2023-11-23
> > **Response to Reviewer HZRy**
> >
> > Dear Reviewer HZRy,
> >
> > Thank you for your detailed review and the valuable feedback. We have carefully provided clarifications and experiments to your previous comments.  We appreciate your thorough evaluation of our work and look forward to hearing from you and addressing any further questions or concerns you may have.
> >
> > Thank you for your continued engagement and support.

---

> ### Author Response · Authors · 2023-11-18
> **Response to Reviewer HZRy (Part 2)**
>
> 4. **The authors state several times that the proposed method improves efficiency. Please demonstrate in detail, e.g., the comparison of inference time.**
>
>     We provide efficiency comparisons in Table F.5, where our model takes less time for inference. We also compare inference speed tested on the A100 GPU and inpainting quality in the table below where our model achieves favorable performance.
>
>     | | | | | | |
>     |---|:---:|:---:|:---:|:---:|:---:|
>     |Method |**Ours**|Stable Diffusion|LDM|ControlNet|MAT|
>     |FID$\downarrow$ | **1.68** | 2.11 | 2.76 | 5.55 | 2.90 |
>     |Speed$\downarrow$ (512×512) |**0.25s**|3.6s|2.7s|3.1s|0.26s|
>     | | | | | | |
>
> 5. **What is the mask size in Table 4?**
>
>     The mask size is 512x512, consistent with the image sizes of Places2 and CelebA-HQ as indicated in Table 4. The small and large mask settings refer to different masking ratios. The mask statistics have been provided in Sec.B of Mat's [e] supplementary file. The small and large masks are established with an average masking ratio of approximately 20% and 50%, respectively.
>
> 6. **In the experiments, please add the comparisons with more recent methods, e.g., [a] Chu et al., Rethinking Fast Fourier Convolution in Image Inpainting, ICCV 2023. [b] Sargsyan et al., MI-GAN: A Simple Baseline for Image Inpainting on Mobile Devices, ICCV 2023. [c] Ko et al., Continuously Masked Transformer for Image Inpainting, ICCV 2023.**
>
>     Thanks for the valuable comments. We have cited all the referred works. In practice, we find out that only MI-GAN [b] provides implementation code and models, so we compare our method to MI-GAN and ControlNet \[d\] (ICCV 2023 Best Paper Award) below. Note that ControlNet is trained on a more extensive LAION dataset rather than the Places2 dataset. The results obtained from our proposed PSM exhibit significant improvements across all metrics. We observe that ControlNet might produce undesired results due to its tendency to generate new objects that might not align harmoniously with the existing content. A detailed comparison is shown in Sec.I due to page limit.
>
>     #### 512x512 Places2 under the small mask ratio setting
>     | | | | | | |
>     |---|---|---|---|---|---|
>     | Method | FID$\downarrow$ | P-IDS$\uparrow$ | U-IDS$\uparrow$ | LPIPS$\downarrow$ | PSNR$\uparrow$ |
>     | **PSM (ours)** | **0.72** | **30.95** | **43.91** | **0.084** | **25.51** |
>     | MIGAN | 1.40 | 18.43 | 39.35 | 0.103 | 24.38 |
>     | ControlNet | 1.86 | 12.63 | 35.71 | 0.117 | 24.68 |
>     | | | | | | |
>
>     #### 512x512 Places2 under the large mask ratio setting
>     | | | | | | |
>     |---|---|---|---|---|---|
>     | Method | FID$\downarrow$ | P-IDS$\uparrow$ | U-IDS$\uparrow$ | LPIPS$\downarrow$ | PSNR$\uparrow$ |
>     |**PSM (ours)**| **1.68** | **25.33** | **39.30** | **0.161** | **20.89** |
>     |MIGAN| 3.81|13.50|32.42|0.195|20.00|
>     |ControlNet|5.55|6.60|25.65|0.219|19.73|
>     | | | | | | |
>
>     [a] Chu, Tianyi, et al. "Rethinking Fast Fourier Convolution in Image Inpainting." ICCV. 2023.
>     [b] Sargsyan, Andranik, et al. "MI-GAN: A Simple Baseline for Image Inpainting on Mobile Devices." ICCV. 2023.
>     [c] Ko, Keunsoo, and Chang-Su Kim. "Continuously Masked Transformer for Image Inpainting." ICCV. 2023.
>     [d] Zhang, Lvmin, Anyi Rao, and Maneesh Agrawala. "Adding conditional control to text-to-image diffusion models." ICCV. 2023.
>     [e] Li, Wenbo, et al. "Mat: Mask-aware transformer for large hole image inpainting." CVPR. 2022.

---

### Author Response · Authors · 2023-11-18
**Response to AC and reviewers**

Dear AC and all reviewers,

We sincerely appreciate your time and efforts in reviewing our paper. We are glad to find that reviewers recognized the following merits of our work:

- **Well-motivated idea [HZRy, EPEv, rxeq]**: Iterative probabilistic algorithms like diffusion models achieve superior results compared to GAN-based methods but often require substantial computing resources. To overcome this limitation, this paper proposes a novel pixel spread model that combines the optimization efficiency of GAN with the prediction tractability of probabilistic models. The idea is novel, and the paper is solid.
- **Impressive performance [HZRy, EPEv, N7bD, rxeq]**: The proposed method demonstrates state-of-the-art performance across multiple benchmarks, competing favorably with recent GAN, transformer, and diffusion models. Notably, the model exhibits robust generalization to unknown mask types and higher resolutions.
- **High efficiency [EPEv, N7bD]**: The method proves to be highly efficient in inpainting high-quality images, largely surpassing the speed of transformer and diffusion methods.

We also thank all reviewers for their insightful and constructive suggestions, which help further improve our paper. In addition to the pointwise responses below, we summarize the major revision in the rebuttal according to the reviewers’ suggestions:

- **Detailed illustration of Eq.1 and iterative process in Sec.3.1.2 [HZRy, EPEv, rxeq]**: We have provided a more comprehensive analysis of each variable in Eq.1. Additionally, we explain step-by-step the uncertainty measure and pixel updating process in Sec.3.1.2 with more details.
- **Additional comparisons to ICCV 2023 methods ([HZRy, EPEv])**: We have tried to collect publicly available methods that appeared in ICCV 2023 for comparison. The comparison results in Places2 dataset show that our method achieves favorable performance.
- **Manuscript update ([HZRy, EPEv, N7bD])**: We have included more in-depth visual comparison analysis, experimental results, and discussions in the main paper and Appendix.

We hope our pointwise responses below can clarify all reviewers' confusion and address the raised concerns. We thank all reviewers' efforts and time again.

Best,
Authors

---

### Meta-Review · Area_Chair_LLni · 2023-12-06

**Metareview:**

The authors propose and iterative approach for image inpainting that can address severe degradations. The proposed technique uses an uncertainty estimate of the pixels, pixel variances for inpainting, to guide the synthesis pipeline. This is done by fusing the input and output of each iteration and guiding the network attention modules.

**Justification For Why Not Higher Score:**

Although the results show significant improvement of quality, the theory behind the proposed technique remains unknown: we do not know if the process converges nor that we know the output is guaranteed to lie on data manifold.

**Justification For Why Not Lower Score:**

The proposed probabilistic pipeline is novel and achieves state-of-the-art results. This approach could motivate further investigation in other problems or theoretical work.

---

### Decision · Program_Chairs · 2024-01-16

Accept (spotlight)